# Comparative Effects of Temporal Interference and High-Definition Transcranial Direct Current Stimulation on Spontaneous Neuronal Activity in the Primary Motor Cortex: A Randomized Crossover Study

**DOI:** 10.3390/brainsci15030317

**Published:** 2025-03-18

**Authors:** Zhiqiang Zhu, Lang Qin, Dongsheng Tang, Zhenyu Qian, Jie Zhuang, Yu Liu

**Affiliations:** 1School of Kinesiology, Shenzhen University, Shenzhen 518000, China; zhuzhiqiang@szu.edu.cn (Z.Z.); qinlang2022@email.szu.edu.cn (L.Q.); tangdongsheng1225@163.com (D.T.); 2Magnetic Resonance Imaging (MRI) Center, Shenzhen University, Shenzhen 518000, China; 3School of Kinesiology, Shanghai University of Sport, Shanghai 200438, China; qianzhenyu@sus.edu.cn (Z.Q.); jie.zhuang@sus.edu.cn (J.Z.)

**Keywords:** temporal interference, non-invasive brain stimulation, dynamic regional homogeneity, resting-state fMRI, primary motor cortex

## Abstract

**Background:** Modulating spontaneous neuronal activity is critical for understanding and potentially treating neurological disorders, yet the comparative effects of different non-invasive brain stimulation techniques remain underexplored. **Objective:** This study aimed to systematically compare the effects of temporal interference (TI) stimulation and high-definition transcranial direct current stimulation (HD-tDCS) on spontaneous neuronal activity in the primary motor cortex. **Methods:** In a randomized, crossover design, forty right-handed participants underwent two 20 min sessions of either TI or HD-tDCS. Resting-state fMRI data were collected at four stages: pre-stimulus baseline (S1), first half of stimulation (S2), second half of stimulation (S3), and post-stimulation (S4). We analyzed changes in regional homogeneity (ReHo), dynamic ReHo (dReHo), fractional amplitude of low-frequency fluctuations (fALFFs), and dynamic fALFFs (dfALFFs) to assess the impact on spontaneous neuronal activity. **Results:** The analysis revealed that TI had a more significant impact on ReHo, especially in the left superior temporal gyrus and postcentral gyrus, compared with HD-tDCS. Both stimulation methods exhibited their strongest effects during the second half of the stimulation period, but only TI maintained significant activity in the post-stimulation phase. Additionally, both TI and HD-tDCS enhanced fALFFs in real-time, with TI showing more pronounced effects in sensorimotor regions. **Conclusions:** These findings suggest that TI exerts a more potent and sustained influence on spontaneous neuronal activity than HD-tDCS. This enhanced understanding of their differential effects provides valuable insights for optimizing non-invasive brain stimulation protocols for therapeutic applications.

## 1. Introduction

Spontaneous neuronal activity, characterized by intrinsic fluctuations in neural firing patterns without external stimuli, plays a fundamental role in shaping brain function and behavior [1]. This activity significantly influences information processing, neuroplasticity, and cognitive function [2]. Modulating spontaneous neuronal activity has emerged as a promising approach in neuroscience research and clinical applications, providing insights into the intrinsic organization of the brain and the mechanisms underlying various cognitive processes [3]. From a clinical perspective, regulating spontaneous neural activity has the potential to treat neurological and psychiatric disorders, such as epilepsy, depression, and Alzheimer’s disease [4,5]. In neurorehabilitation, modulating spontaneous activity can facilitate neural plasticity [6]. Thus, the ability to precisely modulate spontaneous neuronal activity represents a valuable tool across multiple domains, offering unprecedented opportunities to understand brain function, develop innovative treatments, and advance neural engineering applications in the scientific landscape.

Transcranial electrical stimulation has emerged as a noninvasive method for modulating spontaneous neuronal activity, with widespread applications in neuroscience research and clinical interventions [7]. However, traditional transcranial direct current stimulation (tDCS) is limited by its poor spatial resolution and targeting precision. To address these limitations, researchers have developed high-definition tDCS (HD-tDCS), which employs multiple focal electrodes to achieve more precise stimulation targeting [8]. HD-tDCS has been shown to effectively modulate cognitive function and neurophysiological activity [9]. Nevertheless, while HD-tDCS stimulates the deep target regions, it still significantly affects the activity of superficial neural tissues [10]. Recently, temporal interference (TI) stimulation has emerged as a promising approach for overcoming this challenge. TI stimulation utilizes the interference of two different frequency alternating currents to achieve precise stimulation of deep brain regions while minimizing the effects on superficial tissues [11]. Although TI stimulation demonstrates unique advantages, there is still a lack of research on its mechanisms for modulating spontaneous neuronal activity and direct comparisons with traditional transcranial electrical stimulation (tES). Previous studies have shown that 20 Hz transcranial alternating current stimulation (tACS) exhibits significant heterogeneity in regulating motor cortical excitability, which reduces its reliability as a control condition [12,13]. In contrast, Zeng et al. showed that HD-tDCS demonstrated stable electrical field focality properties [14]. With a 4 × 1 ring electrode configuration that improves focusing compared to conventional tDCS, it produces a higher peak polarization of synapses [15]. Therefore, HD-tDCS was selected as the control condition to reveal the unique advantages of the TI. Therefore, investigating the neural modulatory mechanisms of TI stimulation and conducting systematic comparisons with HD-tDCS will provide crucial theoretical foundations and practical guidance for optimizing noninvasive brain stimulation.

Our previous research observed the effects of TI stimulation on primary motor cortex (M1) functional connectivity [16], yet the underlying mechanisms of how TI stimulation influences neuronal spontaneous activity and its dynamic properties remain unclear. Based on this, this study aimed to further investigate the mechanisms of the effects of TI stimulation on the nervous system from the perspective of neuronal spontaneous activity [16]. We hypothesized that these two stimulation methods would produce distinct patterns of modulation of spontaneous neuronal activity within the M1 region [8]. TI stimulation may elicit more widespread alterations in spontaneous activity patterns, potentially affecting deeper cortical layers for its unique mechanism [11]. Furthermore, TI stimulation may result in frequency-specific modulations of neural oscillations, particularly in the gamma band, which has been associated with local circuit interactions and cognitive processing [17]. This comparative investigation may optimize the noninvasive brain stimulation protocols for various neurological and psychiatric applications.

## 2. Materials and Methods

### 2.1. Participants

Subjects participated in our experiment: 9 females (age: 24.11 ± 0.93 years) and 31 males (age: 25.97 ± 3.53 years). The sample size was determined through an a priori power analysis using G*Power 3.1 software [18] to ensure sufficient statistical power for testing the primary hypothesis. The calculation was based on an effect size of f = 0.25, a significance level of α = 0.05, and a power of (1 − β) = 0.80. The required sample size was 34; with an additional 20% dropout rate, the final sample size was 40. The Edinburgh Handedness Scale confirmed that all the participants were right-handed [19]. Participants were selected based on the following criteria: (1) aged 18–35 years; (2) no metal implants, medication history, or prior neurological conditions; and (3) no adverse responses to noninvasive brain stimulation techniques.

Before the commencement of the study, participants were introduced to a preliminary stimulation phase to acclimate to the associated sensations and understand the research procedures. Affirmative written informed consent was obtained from all individuals before their involvement in the experiment. The research was approved by the Shanghai University of Sport (102772020RT116). Additionally, all methods were performed in accordance with the relevant guidelines and regulations and complied with the Declaration of Helsinki.

### 2.2. Experimental Design

Each participant underwent two 20 min sessions of either TI stimulation or HD-tDCS. There was at least a 48 h interval between sessions to wash-out stimulation effects. At each session, participants underwent both functional and structural imaging assessments. The fMRI procedure recorded the brain’s resting-state activity in three phases: a pre-stimulation period of 8 min, a stimulation period of 21 min, and a post-stimulation period of 8 min. After functional imaging, the participants underwent a structural imaging scan that lasted for an additional 6 min (Figure 1). To prevent order effects, the participants were randomly assigned to receive two different types of stimulation by an experimenter who was not involved in data processing.

### 2.3. Brain Stimulation Parameters and Session Procedures

#### 2.3.1. TI

Using transcranial magnetic stimulation (TMS) to target the left primary motor cortex with the lowest possible stimulation intensity. The specific location that induces a visible contraction in the first dorsal interosseous muscle of the left hand upon stimulation is identified as the “target area” [20]. The four electrodes are positioned to form a square with sides that are 4 cm long, with two sides of the square parallel to the line from the glabella (forehead) to the inion (occipital bone), and the other two sides parallel to the line connecting the two ears, surrounding the target point shown in (Figure 2). The R1–R2 channel was 2000 Hz and the L1–L2 channel was 2020 Hz and 2 mA in each channel, the difference in frequencies generated the 20 Hz envelope [21]. TI stimulation was performed using a device from Soterix Medical, Woodbridge, NJ, USA, with a session lasting 20 min that included two brief 30 s ramp-up and ramp-down phases at the beginning and end of the session.

#### 2.3.2. HD-tDCS

Based on previous studies [16], C3 is chosen as the stimulation center using four square MRI-compatible rubber electrodes (1.5 cm × 2 cm) and the same current as for TI stimulation, i.e., maximum 2 mA per channel and calculated the optimal electrode sheet locations using SimNIBS 4.0 by using finite element-based brain stimulation electric fields [22]. The location of each electrode placement and current magnitude, as shown in (Figure 3), were calculated as follows: C3:2000 μA; P3, −774 μA; T7, −684 μA; and Cz, −542 μA. We used the DC-STIMULATOR PLUS (NeuroCnn, Ilmenau, Germany) with a session lasting 20 min that included two brief 30 s ramp-up and ramp-down phases at the beginning and end of the session [23].

### 2.4. Image Acquisition and Preprocessing

Structural and functional images were acquired using a 3.0 Tesla Siemens MAGNETOM Prisma whole-body MRI scanner equipped with a 64-channel head coil for radio frequency (RF) reception (Siemens, Munich, Germany). The 3D MPRAGE sequence was used to obtain T1-weighted images with the following parameters: repetition time (TR) 3130 ms; echo time (TE) 2.98 ms, flip angle, 12°; voxel size, 1 × 1 × 1 mm^3^; and field of view (FOV) 256 × 256 mm^2^. Resting-state fMRI images were collected using an EPI sequence with the following parameters: TR = 1000 ms, TE = 3 ms, FOV = 240 × 240 mm^2^, voxel size = 3 × 3 × 3 mm^3^; three functional runs were performed. The first and third runs consisted of 488 volumes, lasting for 8 min and 8 s, respectively. The second run acquired 1268 brain volumes over 21 s and 8 s. Participants were directed to maintain a state of relaxation with their eyes closed and to remain conscious throughout the procedure. The parameters utilized for both functional and structural imaging were in accordance with the methodology outlined by Zhu et al. [16].

We used a Data Processing Assistant for resting-state fMRI (DPABI, http://rfmri.org/DPABI, accessed on 12 October 2024 [24]; SPM12: http://www.fil.ion.ucl.ac.uk/spm, accessed on 12 October 2024, which are based on MATLAB 2023a) to preprocess the brain imaging data. To maintain consistency across all time points (baseline, during stimulation, and post-stimulation), we employed a standardized duration of 8 min and 8 s for each analysis, analyzing the entire duration for S1 (baseline) and S4 (post-stimulation), while during stimulation, during the 21 min and 8 s stimulation period, we focused our analysis on S2 (the initial 8 min and 8 s) and S3 (the concluding 8 min and 8 s). The initial 10 volumes of data for each participant were discarded to ensure that the signal stabilized to equilibrium levels. Regressions were performed on cerebrospinal fluid and white matter signals, as well as 24 head motion parameters and linear trends, to eliminate the effects of head motion. Head motion was rescaled. Eight participants were excluded from the study due to excessive head motion, with the exclusion criteria set at a maximum displacement of over 2.0 mm or an angular rotation exceeding 2.0°. Functional images were aligned to the MNI space using the DARTEL algorithm, with voxel dimensions standardized to 3 mm × 3 mm × 3 mm [25]. In addition, to minimize low-frequency drift and high-frequency physiological noise, we removed linear trending and bandpass filtering (0.01–0.08 Hz). Spatial smoothing was performed in the final step to improve the signal-to-noise ratio. This was accomplished using an isotropic Gaussian kernel with a full width at half-maximum (FWHM) of 6 mm. It is crucial to note that spatial smoothing should be conducted after calculating the dynamic regional homogeneity (dReHo) index because the smoothing process can influence the regional homogeneity (ReHo) index computation.

### 2.5. Calculation of dReHo and dfALFFs of Resting-State fMRI

Dynamic resting-state fMRI indices were obtained through a sliding time-window analysis using the Data Processing and Analysis of Brain Imaging (DPABI) software (http://rfmri.org/DPABI, accessed on 12 October 2024) [26]. Hamming windows were applied to the resting-state fMRI time series to create a windowed time series. Previous studies have indicated that the optimal window length should range between 10 and 75 TR. In this study, a sliding window approach was employed, with each window encompassing 30 TR, and the window was advanced by 1 TR for each subsequent analysis, yielding a total of 449 distinct windows from the dataset’s 478-time points [27,28]. Subsequently, we performed Z-standardization on the maps across the entire group mask for all voxels to reduce the variability caused by differences in global activity levels among participants. The Coefficient of Variation (CV) maps for both ReHo and fractional amplitude of low-frequency fluctuations (fALFFs) were calculated within each brain window, and individual voxel-wise CV maps were standardized by dividing by the whole-brain mean values and spatially smoothed with a Gaussian kernel with a full-width-at-half-maximum (FWHM) of 6 mm. Additionally, to ensure the stability of the sliding time-window analysis, we repeated all calculations using window widths of 25 and 35 TR.

### 2.6. Statistical Analysis

Functional MRI time series were modeled using a general linear model with SPM12 software (http://www.fil.ion.ucl.ac.uk/spm, accessed on 12 October 2024; Wellcome Trust Centre for Neuroimaging, London, UK) implemented in MATLAB 2020a (MathWorks, Natick, MA, USA). The study employed a two-way repeated ANOVA (2 × 4) to examine the variances in ReHo, dReHo, fALFFs, and dfALFFs between the TI and tDCS groups, considering ‘group’ with two categories (TI and tDCS) and ‘time’ with four stages (pre-stimulus baseline: S1, first stimulus half: S2, second stimulus half: S3, and post-stimulus: S4). We examined the interaction effect between the group and time. Furthermore, a post hoc analysis was conducted for clusters exhibiting significant interaction effects. False discovery rate (FDR) correction with a threshold of *p* < 0.05 was applied at the cluster level to control for multiple comparisons. The coordinates of notable cluster peaks and subpeaks for each effect were documented in tables and presented in standard MNI space. Regions were identified using the AAL atlas and Brodmann areas, as implemented in MRIcron (https://www.nitrc.org/projects/mricron, accessed on 12 October 2024).

## 3. Results

### 3.1. Effect of Different Transcranial Electrical Stimulation on ReHo

After a two-way repeated measures ANOVA, the results showed an interaction effect of stimulation mode and time on the ReHo indicator. Therefore, separate effects were tested for the two within-subject factors (Table 1).

In the second half of the stimulation, the TI group minus the tDCS group showed significant activation in the left superior temporal gyrus and postcentral gyrus (Table 1, Figure 4).

In the tDCS group, the second half of stimulation minus the baseline had a significant activation in the right precentral gyrus, superior temporal gyrus, and left postcentral gyrus brain regions, and the second half of stimulation minus stimulation had a significant activation in the right posterior central gyrus and superior temporal gyrus brain regions (Table 1, Figure 5).

In the TI group, the second half of the stimulus minus the baseline showed significant activation in the right postcentral gyrus, transverse temporal gyrus, and left postcentral gyrus brain regions; the second half of the stimulus minus the first half of the stimulus showed significant activation in the right posterior central gyrus, transverse temporal gyrus, and left superior temporal gyrus brain regions; the second half of the stimulus minus the baseline showed significant activation in the right precentral gyrus brain regions; and the second half of the stimulus minus the post-stimulus showed significant activation in the right posterior central gyrus, superior temporal gyrus, and left transverse temporal gyrus brain regions (Table 1, Figure 5).

### 3.2. Effect of Different Transcranial Electrical Stimulation on dReHo

After a two-way repeated measures ANOVA the results showed an interaction effect of stimulation mode and time for the dReHo indicator. Therefore, separate effects were tested for the two within-subject factors (Table 2).

In the tDCS group, the baseline minus the second half of stimulation showed significant activation in the right superior temporal gyrus, postcentral gyrus, precentral gyrus, medial and paracentral cingulate gyrus, and left transverse temporal gyrus brain regions, and the post-stimulus minus the second half of the stimulus had a significant activation in the right superior temporal gyrus, precentral gyrus, postcentral gyrus, and left transverse temporal gyrus brain regions (Table 2, Figure 6).

In the TI group, the baseline minus the second half of stimulation had a significant activation in the right precentral gyrus, postcentral gyrus, transverse temporal gyrus, left precuneus, postcentral gyrus, superior temporal gyrus, and paracentral lobule brain regions, and the post-stimulus minus the second half of the stimulus had a significant activation in the right superior temporal gyrus, transverse temporal gyrus, insula, and left precentral gyrus brain regions (Table 2, Figure 6).

### 3.3. Effect of Different Transcranial Electrical Stimulation on fALFFs

After a two-way repeated measures ANOVA, the results showed an interaction effect of stimulation mode and time on the fALFF indicator. Therefore, separate effects were tested for the two within-subject factors (Table 3).

In the TI group, the second half of the stimulus minus the baseline showed significant activation in the right posterior central gyrus, transverse temporal gyrus, and left superior temporal gyrus brain regions, and the post-stimulus minus the baseline showed significant activation in the right precentral gyrus, postcentral gyrus, left precentral gyrus, and postcentral gyrus brain regions (Table 3, Figure 7).

### 3.4. Effect of Different Transcranial Electrical Stimulation on dfALFFs

The results of a two-way repeated measures ANOVA using two factors showed no interaction effect between the stimulation method and time. Post hoc analyses of the main effects of the two factors, stimulation method and time, showed no statistically significant effects of either the stimulation method or time factors on the dfALFF values.

## 4. Discussion

The results of this study revealed that tDCS and TI have distinct modulatory effects on brain functional connectivity and low-frequency oscillations, with dynamic changes occurring during and after stimulation. This study employed a 2 × 4 design to compare the effects of HD-tDCS and TI on brain function across four time points (baseline S1, first half of stimulation S2, second half of stimulation S3, and post-stimulation S4). We examined changes in ReHo, dReHo, fALFFs, and dfALFFs. The results revealed significant differences in ReHo between the TI and HD-tDCS groups during S3, particularly in the left superior temporal gyrus and postcentral gyrus. Both groups showed significant within-group differences across time points in ReHo and dReHo, involving areas such as the precentral, postcentral, superior temporal, and transverse temporal gyri. The fALFF analysis showed significant activation in the TI group, notably between S3 and S1 and between S4 and S1, in regions including the postcentral gyrus, transverse temporal gyrus, and superior temporal gyrus. No significant effects were observed for the dfALFFs.

### 4.1. ReHo and dReHo

Our study revealed significant increases in ReHo in the left superior temporal gyrus and postcentral gyrus at S3 in TI compared with HD-tDCS. Intra-group comparisons also showed significant time effects for both stimulations, involving brain regions such as the precentral gyrus, postcentral gyrus, superior temporal gyrus, and transverse temporal gyrus. Both types of stimulation had the strongest effect at S3, while TI showed significant activation in S4 compared to S1. The dReHo analysis indicated that there was a significant decrease in activation at S3 for both stimulations compared to S1 and S4. This decrease primarily involves similar brain areas.

Our findings indicate that both TI and HD-tDCS significantly impact regional brain functional connectivity and local activity, albeit with different patterns of effects. At S3, TI induced a more pronounced increase in ReHo in the left superior temporal gyrus and postcentral gyrus than HD-tDCS. This suggests that TI may have a stronger modulatory effect on local functional synchronization in these regions [29]. Both stimulation types exhibited their strongest effects at S3, involving the sensorimotor cortex (precentral and postcentral gyri). These findings imply that transcranial electrical stimulation may influence brain activity by modulating the function of sensorimotor networks [30]. This influence is particularly noticeable in the later part of the stimulus. Interestingly, TI showed significant activation at S4 compared with S1, potentially indicating a more sustained modulatory effect. However, dReHo analysis revealed a significant decrease in activation at S3 for both stimulation types. This result indicates that, in the later part of the stimulus, there was a decrease in the variability of regional brain functional connectivity within the sensorimotor cortices, leading to a more stable regional brain functional connection.

Our results align with previous findings in some respects while providing new insights. Previous studies have demonstrated that HD-tDCS can modulate regional brain functional connectivity and activity [31]. Our results further confirm this and extend our understanding of TI effects. We found that TI produced stronger modulatory effects in certain brain regions than HD-tDCS [21].

Moreover, our study revealed some novel findings. We observed that both stimulation types had the strongest effect at S3. The online effects of the two types of stimulation exhibit time dependency, as the stimulus effect strengthens with longer stimulus duration [32]. Moreover, to the best of our knowledge, this is the first study to observe the effects of tES on dReHo. At S3, two different types of electrical stimulation led to an increase in the consistency of activity within the local brain regions, making it more stable. Such an enhancement may be more effective in boosting brain function within specific areas [33].

Overall, our research results not only support the view that transcranial electrical stimulation can effectively modulate brain function but also provide new insights into the differential effects of various stimulation methods (HD-tDCS vs. TI) and the dynamic characteristics of stimulation effects over time. These findings provide an important basis for the further optimization of transcranial electrical stimulation protocols and a deeper understanding of their neuromodulatory mechanisms.

### 4.2. fALFFs and dfALFFs

Our analysis of the fALFFs revealed significant activation in the TI, particularly in the right postcentral gyrus, transverse temporal gyrus, and left superior temporal gyrus during S3 compared to S1. Additionally, S4 showed significant activation in the bilateral precentral and postcentral gyri compared with S1. No significant effects were observed for the dfALFFs in either group.

fALFFs, which reflect the relative contribution of specific low-frequency oscillations to the entire detectable frequency range, are thought to indicate the intensity of regional spontaneous brain activity [34]. The observed increases in fALFFs during and after TI stimulation suggest that TI may enhance spontaneous neuronal activity in specific brain regions, potentially through the modulation of low-frequency oscillations. This finding is consistent with previous research showing that transcranial alternating current stimulation (tACS) can modulate spontaneous low-frequency oscillations, which are thought to reflect the intensity of regional spontaneous brain activity [35]. This consistency highlights the potential of TI to modulate spontaneous brain activity.

However, our study also found some discrepancies with previous studies. For example, a study on aging and cognition found that a decreased amplitude of low-frequency fluctuations (ALFFs) was associated with cognitive decline in older adults [36]. Similarly, a study on schizophrenia found that patients with schizophrenia showed decreased ALFFs in the prefrontal and anterior cingulate cortices, which are related to their cognitive deficits [37]. These findings suggest that a decreased ALFF may be related to impaired cognitive function. In contrast, our study found that TI stimulation increased the fALFFs in motor-related cortical areas, which may have implications for improving motor function by modulating sensory processing and motor control networks.

The lack of significant dfALFF results indicates that the dynamic changes in low-frequency fluctuations may not be as pronounced or consistent as the static changes captured by the fALFFs. This suggests that TI or HD-tDCS has no effect on the variation in fALFFs.

### 4.3. Mechanisms

The results of this study provide new insights into the neurophysiological mechanisms underlying tES. The observed changes in ReHo, dReHo, and fALFFs suggest that these stimulations may exert their effects by modulating the local neural synchrony and spontaneous brain activity.

TI demonstrated superior efficacy in modulating local brain functional connectivity compared to HD-tDCS. Our study revealed that TI stimulation not only exerted stronger stimulation effects but also had a longer duration. This may be attributed to the better penetrability and focality of TI stimulation, which allows for more focality stimulation [11]. Previous studies have consistently shown that TI stimulation possesses superior focality and penetrability [21,38]. When examining the dynamic changes in local brain functional connectivity, both TI- and HD-tDCS reduced ReHo variability, with no difference between the two stimulation types. This may be due to both stimulations increasing the ReHo to its extremum value, resulting in reduced ReHo fluctuations.

Our study demonstrates that both TI- and HD-tDCS can enhance fALFFs in online stimulation, suggesting that these stimulation techniques may operate through multiple mechanisms. First, the enhancement of fALFFs indicates that both TI and HD-tDCS successfully increased neural activity in the target brain regions, possibly by altering neuronal membrane potentials, thereby enhancing local spontaneous neural activity [39]. Second, the fALFF enhancement induced by stimulation may reflect changes in neuroplasticity. Stagg et al. showed that HD-tDCS can modulate brain excitability and plasticity by regulating GABAergic neurotransmission, and TI may promote neuroplasticity through similar mechanisms [40]. Third, changes in fALFFs may influence broader functional networks. Polanía et al. found that tDCS can modulate functional connectivity in large-scale brain networks, particularly between the prefrontal and parietal cortices, and TI may possess similar network modulation capabilities [41]. Furthermore, Zheng et al. demonstrated that tDCS can increase local cerebral blood flow and glucose metabolism, which could be potential mechanisms underlying fALFF enhancement. TI may similarly affect energy metabolism [42]. Lastly, Meinzer et al. found that tDCS-induced changes in brain activity correlated with improvements in cognitive functions, such as word retrieval, suggesting that future research should investigate whether fALFF changes induced by TI and HD-tDCS can predict improvements in specific cognitive or behavioral functions [43].

In the comparison of tES, tDCS employs a 1–2 mA weak direct current to directionally modulate brain region activity: the anode enhances neuronal excitability to promote regional functionality, while the cathode inhibits neuronal activity, achieving sustained improvements in neural circuit excitability and motor function [44]. tACS utilizes sinusoidal electric fields with periodic oscillations to forcibly reorganize the firing rhythms of neuronal populations and to promote phase synchronization between neuronal discharges and external stimuli. Applying a 20 Hz alternating current synchronized with the brain’s beta rhythm (13–30 Hz) enables targeted modulation of intrinsic neural oscillation patterns in motor control-related brain regions (e.g., the motor cortex), optimizing motor function [45]. However, both stimulation modalities showed limited electric-field focality. To overcome the limitations of focality, TI stimulation utilizes two slightly different high-frequency currents (2000 Hz and 2020 Hz) to generate a low-frequency (20 Hz) envelope that can penetrate deep brain regions and achieve more focal stimulation. Similar to tACS, TI modulates intrinsic neural oscillations in the brain to enhance motor function [46]. These techniques provide valuable tools for exploring the relationships between local neural activity and specific cognitive or behavioral functions [47].

These findings highlight the translational potential of TI and HD-tDCS in clinical applications. Both techniques demonstrate significant modulation of brain functional connectivity and spontaneous activity, which are essential for improving motor and cognitive functions [48]. In stroke rehabilitation, TI’s stronger and more sustained effects of TI on regions, such as the postcentral and superior temporal gyri, suggest its potential to enhance neuroplasticity in sensorimotor networks. This can accelerate the recovery of motor function and sensory integration [49]. Similarly, modulation of cortical excitability by HD-tDCS may complement traditional therapies by reinforcing the neural circuits responsible for motor control [50]. For neuropsychiatric disorders, TI and HD-tDCS have shown the ability to alter low-frequency oscillations and strengthen local synchrony, mechanisms linked to mood regulation and cognitive improvement [51]. TI’s superior focality and deeper penetration might allow better targeting of dysfunctional brain areas, such as the basal ganglia, implicated in Parkinson’s and Huntington’s diseases [52].

### 4.4. Limitations and Future Directions

This study has several limitations that should be addressed in future research and includes the following: (1) Lack of behavioral data. This study lacks assessments of cognitive and motor function, which restricts the direct translation of neurophysiological findings to real-world applications. Future studies could systematically integrate behavioral metrics (e.g., cognitive testing batteries and motor performance evaluations) to address this limitation. (2) Sample characteristics. A small sample size and relatively uneven gender distribution may limit the generalizability of our findings to broader populations. (3) Individual differences. Individual differences, such as skull thickness, may significantly influence the stimulating effects of TI. Future studies should investigate personalized stimulation protocols to optimize the efficacy of TI. (4) Lack of sham conditions and proprioception data. Without the sham condition, the potential confounding influence of scalp sensations associated with active stimulation cannot be excluded, and the experimental design did not isolate the contribution of high-frequency carrier signals from the intended low-frequency envelope modulation. To address both limitations, future studies should implement a control sham condition using two identical high-frequency carriers (e.g., 2000 Hz) to exclude the effects of high-frequency stimulation. (5) Lack of long-term effects. The current study did not investigate whether the observed changes in spontaneous neuronal activity persisted beyond the stimulation period. Future studies could include longer follow-up periods and repeated measurements to determine the duration and stability of these effects. (6) Target areas differ. The locations of the stimulation target areas differed between TI and HD-tDCS, which may be one of the reasons for the discrepancies in the research results.

## 5. Conclusions

This study assessed the effects of TI and HD-tDCS on spontaneous neuronal activity. Key findings include the following: (1) TI has a stronger modulatory effect on regional homogeneity, especially in the left superior temporal gyrus and postcentral gyrus. (2) TI’s post effects on regional homogeneity were more sustained than HD-tDCS. (3) Both TI and HD-tDCS effectively enhanced the amplitude of local low-frequency fluctuations in real-time. Overall, TI appears to exert a more potent and enduring influence on spontaneous neuronal activity than HD-tDCS. These findings contribute to our understanding of neuromodulation techniques and may inform future therapeutic applications of brain stimulation.

## Figures and Tables

**Figure 1 brainsci-15-00317-f001:**
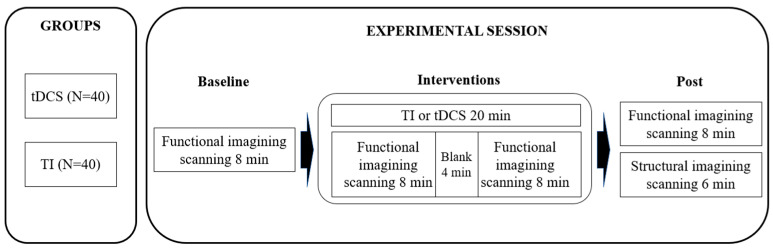
Experimental Design. TI, temporal interference stimulation; tDCS, transcranial direct current stimulation.

**Figure 2 brainsci-15-00317-f002:**
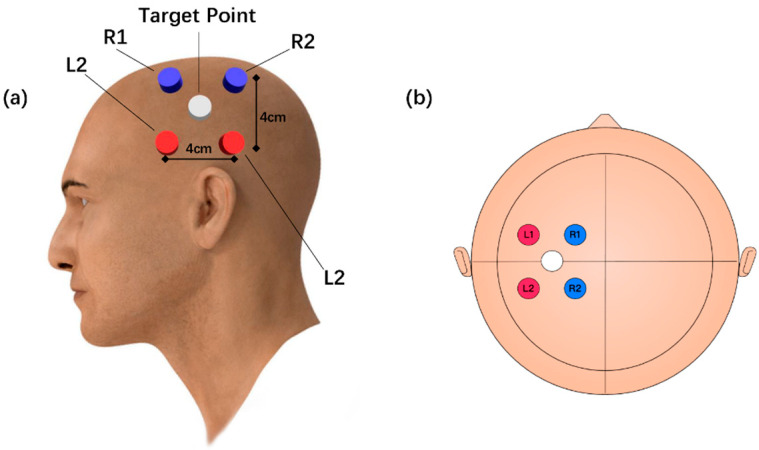
Simulation model, electrode placement of temporal interference stimulation (**a**) Example of electrode position. (**b**) Electrode position in 10–20 system. The white areas are the stimulus target points identified using transcranial magnetic stimulation. Blue and red for the position of the electrodes, two electrodes for a channel, the blue channel frequency of 2000 Hz, and the red channel frequency of 2020 Hz, where the current size is 2 mA. The 4 electrodes form a square with a side length of 4 cm centered on the target point.

**Figure 3 brainsci-15-00317-f003:**
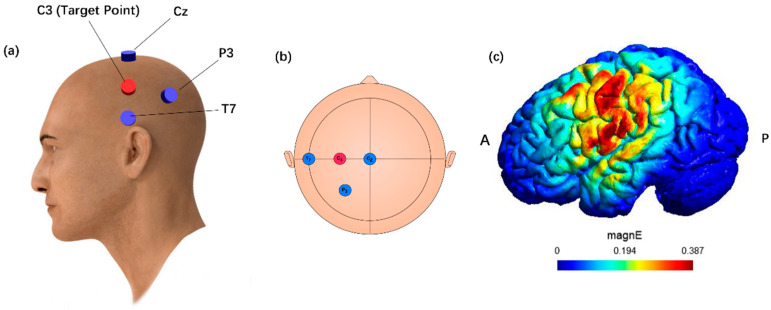
Simulation model, electrode placement, and electrical field of high-definition transcranial direct current stimulation (**a**) Example of electrode position. The blue and red colors show where the cathode and anode of the electrode are placed, respectively. (**b**) Electrode position in 10–20 system. Anode (C3, 2000 μA), Cathodes (T7, −684 μA; P3, −774 μA; Cz, −542 μA). (**c**) Electric field simulation diagram. A: anterior; P: posterior.

**Figure 4 brainsci-15-00317-f004:**
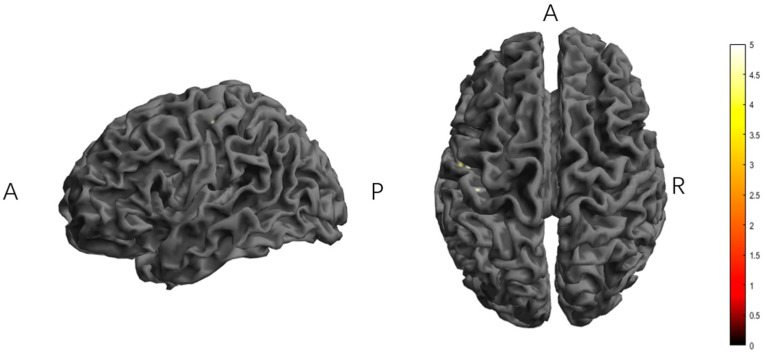
The significant differences in brain regions of ReHo between tDCS-S3 and TI-S3. Note: the color bar represents the T value. S3, the second half of the stimulation; ReHo, regional homogeneity; TI, temporal interference stimulation; tDCS, transcranial direct current stimulation; A, anterior; R, right; P, posterior. Report results with edge-level *p* < 0.001 and cluster-level *p* < 0.05.

**Figure 5 brainsci-15-00317-f005:**
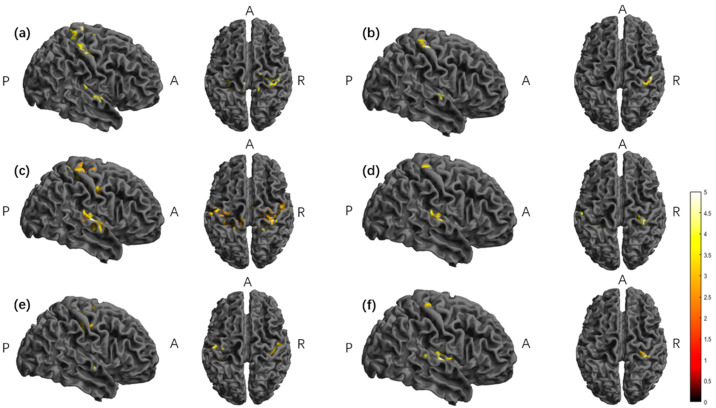
The significant differences in brain regions of ReHo. (**a**) Between tDCS-S3 and tDCS-S1. (**b**) Between tDCS-S3 and tDCS-S4. (**c**) Between TI-S3 and TI-S1. (**d**) Between TI-S3 and TI-S2. (**e**) Between TI-S4 and TI-S1. (**f**) Between TI-S3 and TI-S4. Note: the color bar represents the T value; S1: baseline; S2: first half of the stimulus; S3: second half of the stimulus; S4: post-stimulus; TI, temporal interference stimulation; tDCS, transcranial direct current stimulation; A, anterior; R, right; P, posterior. ReHo, regional homogeneity; report results with edge-level *p* < 0.001 and cluster-level *p* < 0.05.

**Figure 6 brainsci-15-00317-f006:**
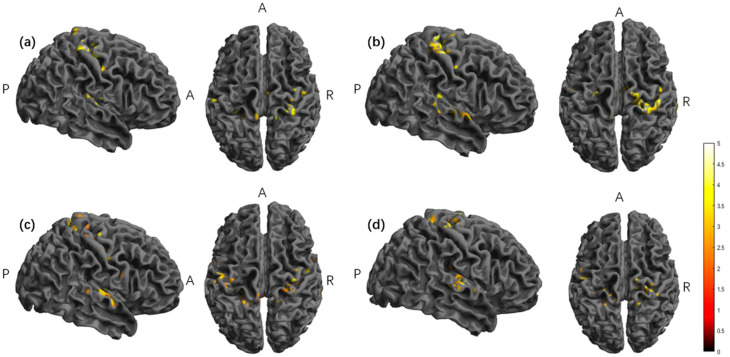
The significant differences in brain regions of dReHo. (**a**) Between tDCS-S1 and tDCS-S3. (**b**) Between tDCS-S4 and tDCS-S3. (**c**) Between TI-S1 and TI-S3. (**d**) Between TI-S4 and TI-S3. Note: The color bar represents the T value. S1: baseline; S3: second half of the stimulus; S4: post-stimulus; dReHo, dynamic ReHo; TI, Temporal Interference stimulation; tDCS, transcranial Direct Current Stimulation; A, anterior; R, right; P, posterior. Report results with edge-level *p* < 0.001 and cluster-level *p* < 0.05.

**Figure 7 brainsci-15-00317-f007:**
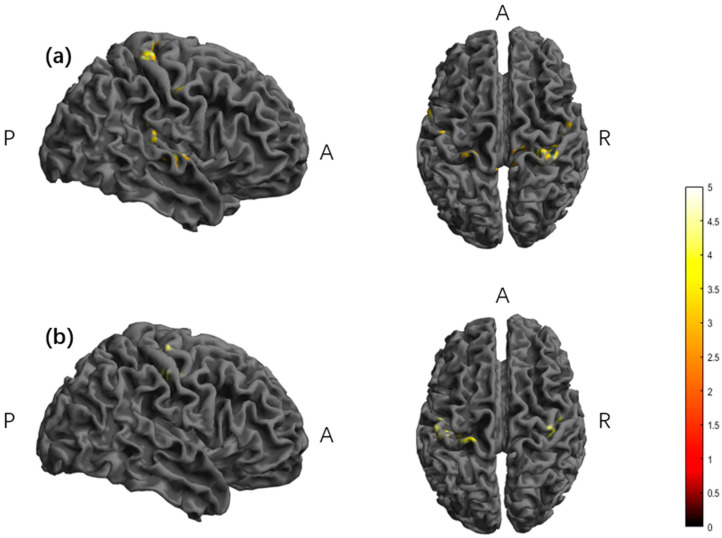
The significant differences in brain regions of fALFFs. (**a**) Between TI-S3 and TI-S1. (**b**) Between TI-S4 and TI-S1. Note: The color bar represents the T value. S1: baseline; S3: second half of the stimulus; S4: post-stimulus; TI, temporal interference stimulation; A, anterior; R, right; P, posterior. fALFFs, fractional amplitude of low-frequency fluctuations; Report results with edge-level *p* < 0.001 and cluster-level *p* < 0.05.

**Table 1 brainsci-15-00317-t001:** Significant differences in ReHo.

Comparisons	Brain Regions/BA	Peak MNI Coordinates			Cluster Voxels	Peak t Values
		x	y	z		
Interaction effect group × time	Temporal_Sup_R	54	−12	3	46	4.57
	Insula_R	36	−15	15	20	4.21
	Postcentral_R	36	−33	63	18	4.09
TI S3–tDCS S3	Temporal_Sup_L	−48	−12	−3	26	4.53
	Postcentral_L	−45	−18	57	17	3.66
tDCS S3–tDCS S1	Precentral_R	39	−27	60	51	4.32
	Temporal_Sup_R	54	−9	0	24	4.04
	Postcentral_L	−27	−39	66	34	3.86
tDCS S3–tDCS S4	Postcentral_R	45	−27	60	20	4.74
	Temporal_Sup_R	54	−12	3	20	4.17
TI S3–TI S1	Heschl_R	48	−21	6	204	5.48
	Postcentral_R	30	−33	63	158	5.38
	Postcentral_R	−48	−24	9	155	5.09
	Postcentral_L	−45	−18	54	91	4.63
	Postcentral_R	24	−45	69	25	4.51
	Postcentral_L	−27	−39	60	92	4.23
TI S3–TI S2	Temporal_Sup_L	−48	−15	−3	57	5.40
	Heschl_R	48	−21	6	36	5.05
	Postcentral_R	30	−33	60	26	4.49
TI S4–TI S1	Precentral_R	51	−18	45	68	4.73
TI S3–TI S4	Temporal_Sup_R	60	−33	9	20	4.86
	Heschl_L	−45	−15	6	69	4.33
	Postcentral_R	33	−33	63	24	4.32

Notes. BA, Brodmann’s area; L, left; R, right; S1, baseline; S2, first half of the stimulus; S3, second half of the stimulus; S4, post-stimulus; TI, temporal interference stimulation; tDCS, transcranial direct current stimulation; ReHo, regional homogeneity; interaction, stimulation type × time interaction effect assessed via two-way repeated measures ANOVA. Report results with edge-level *p* < 0.001 and cluster-level *p* < 0.05.

**Table 2 brainsci-15-00317-t002:** Significant differences in dReHo.

Comparisons	Brain Regions/BA	Peak MNI Coordinates			Cluster Voxels	Peak t Values
		x	y	z		
Interaction effect group × time	Heschl_L	−45	−15	6	20	4.76
	Postcentral_R	33	−30	66	63	4.58
	Heschl_R	45	−21	6	31	4.37
tDCS S1–tDCS S3	Temporal_Sup_R	54	−18	6	13	4.59
	Postcentral_R	33	−33	66	26	4.46
	Cingulate_Mid_R	9	−15	45	27	4.43
	Heschl_L	−45	−15	6	15	4.29
	Precentral_R	48	−15	39	15	3.72
tDCS S4–tDCS S3	Heschl_L	−45	−15	6	18	4.92
	Precentral_R	30	−30	66	91	4.74
	Postcentral_R	45	−12	33	18	4.60
	Temporal_Sup_R	57	−9	3	19	3.84
TI S1–TI S3	Heschl_R	45	−21	6	81	5.34
	Precuneus_L	−9	−45	63	44	5.16
	Postcentral_L	−45	−18	51	56	4.92
	Postcentral_R	12	−39	72	10	4.76
	Heschl_R	−42	−27	9	31	4.70
	Temporal_Sup_L	−54	−9	3	24	4.54
	Postcentral_R	27	−35	60	16	4.44
	Paracentral_Lobule_L	−6	−36	72	18	4.37
	Precentral_R	33	−21	57	12	4.17
TI S4–TI S3	Supp_Motor_Area_R	6	−3	45	44	4.96
	Temporal_Sup_R	60	−36	9	31	4.90
	Temporal_Sup_R	−33	−27	12	57	4.54
	Insula_R	36	−15	6	11	4.52
	Heschl_R	48	−18	9	31	4.42
	Precentral_L	−33	−27	54	12	3.43

Notes. BA, Brodmann’s area; L, left; R, right; S1, baseline; S2, first half of the stimulus; S3, second half of the stimulus; S4, post-stimulus; TI, temporal interference stimulation; tDCS, transcranial direct current stimulation. dReHo, dynamic, regional homogeneity; interaction, stimulation type× time interaction effect assessed via two-way repeated measures ANOVA. Report results with edge-level *p* < 0.001 and cluster-level *p* < 0.05.

**Table 3 brainsci-15-00317-t003:** Significant differences in fALFFs.

Comparisons	Brain Regions/BA	Peak MNI Coordinates			Cluster Voxels	Peak t Values
		x	y	z		
Interaction effect group × time	Temporal_Sup_R	51	−12	0	10	3.89
TI S3–TI S1	Temporal_Sup_L	−51	−12	0	9	4.70
	Postcentral_R	30	−36	63	45	4.37
	Postcentral_R	24	−42	72	8	4.36
	Heschl_R	54	−6	3	16	4.27
TI S4–TI S1	Postcentral_R	48	−21	48	6	4.53
	Precentral_R	42	−18	45	15	4.06
	Postcentral_L	−48	−24	51	7	4.06
	Precentral_L	−27	−27	72	8	3.89
	Postcentral_L	−33	−45	66	7	3.71

Notes. BA, Brodmann’s area; L, left; S1, baseline; S2, first half of the stimulus; S3, second half of the stimulus; S4, post-stimulus; TI, temporal interference stimulation. fALFFs, fractional amplitude of low-frequency fluctuations; Interaction, stimulation type× time interaction effect assessed via two-way repeated measures ANOVA. Report results with edge-level *p* < 0.001 and cluster-level *p* < 0.05.

## Data Availability

The original contributions presented in this study are included in the article/Appendix A. Further inquiries can be directed to the author (e-mail: zhuzhiqiang@szu.edu.cn).

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
