# Peer review of "Comparative Effects of Temporal Interference and High-Definition Transcranial Direct Current Stimulation on Spontaneous Neuronal Activity in the Primary Motor Cortex: A Randomized Crossover Study"

_brainsci, 2025, doi:10.3390/brainsci15030317_

Round 1
Reviewer 1 Report
Comments and Suggestions for Authors
Zhu and colleagues compare the effects of TI and HD tDCS when applied over the motor cortex using pre, post and concurrent fMRI. Specifically, they report on differences in spontaneous neuronal activity in response to the respective tES protocol, while in a previous publication of theirs on the same dataset they have reported on the differences in functional connectivity (https://pubmed.ncbi.nlm.nih.gov/35140781/). However, there are several key points that would need to be clarified.
Major comments
- You have previously reported other results from the same dataset (https://pubmed.ncbi.nlm.nih.gov/35140781/): please be clear on this. You could discuss how your current data complements the previously reported observations.
- I don’t fully understand why this comparison was done in the first place. Surely, if you’d like to compare HD-tES to TI, you’d specifically be interested in comparing TI to either HD-tACS or perhaps oscillatory tDCS with the same frequency (in this case 20 Hz HD-tACS or 20 Hz otDCS)? Here, you would need to 1) in the introduction section explain why this particular comparison would be informative, 2) in the discussion section put your current findings in the context of previous findings on the differences found simply between tACS vs tDCS on their effect when applied over the motor cortex. What effects here are simply attributable to differences between tASC vs tDCS, and what is specific for TI, can you state this, is this comparison informative about TI-specific effects?
- Also, why was there no sham condition? This makes it particularly challenging to understand what was driving the effects.
- You write that you targeted the left M1 with TMS to induce a response in the left FDI. This should be impossible. You mean right FDI?
- Do I understand correctly that the placement of the stimulating electrodes for TI and HD-tDCS differed? For TI, you placed the electrodes around the M1 hotspot identified with TMS, while for the HD-tDCS you placed them around the 10-20 position C3. Why, and could you please provide the distance between the hotspot target and the C3 location per participant?
Minor comments
- The references 12 and 13 are the same, and 13 does not relate to the statement concerning your expected effect of TI in the gamma band.
- The age of your females and males are identical, this must be a typo.
- There is a lack of any specific references in your methods sections describing your tDCS and TI protocols respectively, please reference prior work.
I would advise the authors to have the manuscript language edited and proof read by a native English speaker.
Author Response
Response to Reviewer 1 Comments
We would like to sincerely thank the reviewers for their helpful recommendations. We have seriously considered all the comments and carefully revised the manuscript accordingly. All revisions are marked using track changes in the revised document for review. In addition, we have enhanced the language of the manuscripts. We feel that the quality of the manuscript has been significantly improved with these modifications and improvements based on the reviewers’ suggestions and comments. We hope our revision will lead to an acceptance of our manuscript for publication in the Brain Science.
General comment
Zhu and colleagues compare the effects of TI and HD tDCS when applied over the motor cortex using pre, post and concurrent fMRI. Specifically, they report on differences in spontaneous neuronal activity in response to the respective tES protocol, while in a previous publication of theirs on the same dataset they have reported on the differences in functional connectivity (https://pubmed.ncbi.nlm.nih.gov/35140781/). However, there are several key points that would need to be clarified.
Respond to GC 1:
Thank you very much for providing us with numerous professional review comments. As your comments, we have revised the paper accordingly. With your help, the quality of our manuscript has been significantly enhanced. Again, we sincerely appreciate your invaluable help.
Major comment 1. You have previously reported other results from the same dataset ( https://pubmed.ncbi.nlm.nih.gov/35140781/ ): please be clear on this. You could discuss how your current data complements the previously reported observations.
Respond to MC 1:
Thank you for your suggestion. We previously reported related findings on functional connectivity from the same dataset. In the current study, we further analyzed the data from a novel perspective, focusing on the spontaneous activity of neurons and its dynamic aspects. Compared to our earlier functional connectivity studies, we not only examined the spontaneous activities of neurons but also investigated them from the perspective of dynamic changes. To further elucidating the innovative perspective of this research, we added the following contents in the Introduction section as follow.
“Our previous research observed the effects of TI stimulation on primary motor cortex (M1) functional connectivity [1], yet the underlying mechanisms of how TI stimulation influences neuronal spontaneous activity and its dynamic properties remain unclear. Based on this, the study aimed to further investigate the mechanisms of the effects of TI stimulation on the nervous system from the perspective of neuronal spontaneous activity. (P2, line 76-80)
Reference:
[1] Zhu, Z.; Xiong, Y.; Chen, Y.; Jiang, Y.; Qian, Z.; Lu, J.; Liu, Y.; Zhuang, J. Temporal Interference (TI) Stimulation Boosts Functional Connectivity in Human Motor Cortex: A Comparison Study with Transcranial Direct Current Stimulation (tDCS). Neural Plast 2022, 2022, 7605046, doi:10.1155/2022/7605046.
Major comment 2. I don’t fully understand why this comparison was done in the first place. Surely, if you’d like to compare HD-tES to TI, you’d specifically be interested in comparing TI to either HD-tACS or perhaps oscillatory tDCS with the same frequency (in this case 20 Hz HD-tACS or 20 Hz otDCS)? Here, you would need to 1) in the introduction section explain why this particular comparison would be informative, 2) in the discussion section put your current findings in the context of previous findings on the differences found simply between tACS vs tDCS on their effect when applied over the motor cortex. What effects here are simply attributable to differences between tACS vs tDCS, and what is specific for TI, can you state this, is this comparison informative about TI-specific effects?
Respond to MC 2:
We appreciate your insightful comments and suggestions regarding the experimental design. As you suggested, we have carefully revised our manuscript to clarify the reason for the comparison between TI and HD-tDCS in the introduction section, and discuss the TI-specific mechanisms in the discussion section.
- Comparison Rationale: Given the inconsistent results reported in previous studies on 20 Hz tACS, we opted for HD-tDCS as a more reliable control condition. This comparison allows us to better assess the unique advantages of TI in terms of focality and minimizing effects on brain tissues. To clarify this issue, we have added the following content in the introduction section:
“Previous studies have shown that 20 Hz tACS exhibits significant heterogeneity in regulating motor cortical excitability [1-2], which reduces its reliability as a control condition. In contrast, HD-tDCS has demonstrated stable electrofield focusing properties [3]. With a 4×1 ring electrode configuration that improves focusing compared to conventional tDCS, it produces higher peak polarization of synapses [4]. Therefore, this study selected HD-tDCS as the control condition to reveal the unique focusing advantages of TI. (P2, line 65-72)
- Discussion of TI Mechanisms: We have substantially revised the Discussion section to clarify tACS/tDCS inherent differences from TI-specific effects and added the content in the introduction section as follows:
“In the comparison of tES, tDCS employs 1-2 mA weak direct current to directionally modulate brain region activity: the anode enhances neuronal excitability to promote regional functionality, while the cathode inhibits neuronal activity, achieving sustained improvements in neural circuit excitability and motor function [5]. tACS utilizes sinusoidal electric fields with periodic oscillations to forcibly reorganize the firing rhythms of neuronal populations, promoting phase synchronization between neuronal discharges and external stimuli. By applying 20 Hz alternating current synchronized with the brain's beta rhythm (13-30 Hz), it enables targeted modulation of intrinsic neural oscillation patterns in motor control-related brain regions (e.g., the motor cortex), optimizing motor function [6]. However, both stimulation modalities show limited electric field focality. To overcome the limitations of focality, TI stimulation utilizes two slightly frequency different high-frequency currents (2000Hz and 2020Hz) to generate a low-frequency (20Hz) envelope. As brain tissue does not respond to high-frequency alternating current stimulation but only to low-frequency envelope electric fields, TI can achieve focused stimulation in deep brain regions. The same to tACS, TI modulate brain intrinsic neural oscillation to enhance motor function [7].” (P15, line 444-457)
Reference:
[1] Suzuki, M., Tanaka, S., Gomez-Tames, J., Okabe, T., Cho, K., Iso, N., & Hirata, A. (2022). Nonequivalent After-Effects of Alternating Current Stimulation on Motor Cortex Oscillation and Inhibition: Simulation and Experimental Study. Brain sciences, 12(2), 195. https://doi.org/10.3390/brainsci12020195
[2] Wang, L., Nitsche, M. A., Zschorlich, V. R., Liu, H., Kong, Z., & Qi, F. (2021). 20 Hz Transcranial Alternating Current Stimulation Inhibits Observation-Execution-Related Motor Cortex Excitability. Journal of personalized medicine, 11(10), 979. https://doi.org/10.3390/jpm11100979
[3] Zeng, Y., Cheng, R., Zhang, L., Fang, S., Zhang, S., Wang, M., Lv, Q., Dai, Y., Gong, X., & Liang, F. (2024). Clinical Comparison between HD-tDCS and tDCS for Improving Upper Limb Motor Function: A Randomized, Double-Blinded, Sham-Controlled Trial. Neural plasticity, 2024, 2512796. https://doi.org/10.1155/2024/2512796
[4] Aberra, A. S., Wang, R., Grill, W. M., & Peterchev, A. V. (2023). Multi-scale model of axonal and dendritic polarization by transcranial direct current stimulation in realistic head geometry. Brain stimulation, 16(6), 1776–1791. https://doi.org/10.1016/j.brs.2023.11.018
[5] Pelletier, S. J., & Cicchetti, F. (2014). Cellular and molecular mechanisms of action of transcranial direct current stimulation: evidence from in vitro and in vivo models. The international journal of neuropsychopharmacology, 18(2), pyu047. https://doi.org/10.1093/ijnp/pyu047
[6] Ma, R., Xia, X., Zhang, W., Lu, Z., Wu, Q., Cui, J., Song, H., Fan, C., Chen, X., Zha, R., Wei, J., Ji, G. J., Wang, X., Qiu, B., & Zhang, X. (2022). High Gamma and Beta Temporal Interference Stimulation in the Human Motor Cortex Improves Motor Functions. Frontiers in neuroscience, 15, 800436. https://doi.org/10.3389/fnins.2021.800436
[7] Zhu, Z., & Yin, L. (2023). A mini-review: recent advancements in temporal interference stimulation in modulating brain function and behavior. Frontiers in human neuroscience, 17, 1266753. https://doi.org/10.3389/fnhum.2023.1266753
Major comment 3. Also, why was there no sham condition? This makes it particularly challenging to understand what was driving the effects.
Respond to MC 3:
We sincerely appreciate your insightful suggestion regarding the experimental design. Due to funding limitations, we prioritized validating the novel TI protocol against the well-established HD-tDCS approach. We agree that the absence of a sham stimulation condition is a significant limitation of this study. In response to your critical comments, we have incorporated limitation in the revised manuscript. The added content is as follow:
“4) lack of sham condition and proprioception data. Without sham condition, the potential confounding influence of scalp sensation associated with active stimulation cannot be excluded and the experimental design did not isolate the contribution of high-frequency carrier signals from the intended low-frequency envelope modulation. To address both limitations, future studies should implement a control sham condition: The sham condition using two identical high-frequency carriers (e.g., 2000 Hz) to exclude the effects of high-frequency stimulation.” (P16, line 483-495)
Major comment 4. You write that you targeted the left M1 with TMS to induce a response in the left FDI. This should be impossible. You mean right FDI?
Respond to MC 4:
Thank you very much for your attention to the detail of our manuscript. We appreciate your pointing out the error in our manuscript. We have corrected the typo immediately to ensure that the description accurately reflects our experimental design. Specifically, we meant to indicate that targeting the left M1 should induce a response in the right FDI.
“The specific location that induces a visible contraction in the first dorsal interosseous muscle of the right hand upon stimulation is identified as the "target area".” (P3, line 127-129
Major comment 5. Do I understand correctly that the placement of the stimulating electrodes for TI and HD-tDCS differed? For TI, you placed the electrodes around the M1 hotspot identified with TMS, while for the HD-tDCS you placed them around the 10-20 position C3. Why, and could you please provide the distance between the hotspot target and the C3 location per participant?
Respond to MC 5:
Thank you very much for your comments. You are right. The target of TI stimulation is the M1 hotspot identified by TMS, while HD-tDCS involves placing electrodes at the C3 position of the 10-20 system. This placement was based on previous study, which indicated that the C3 on the scalp location is corresponds with the right-hand area, often referred to as the ‘hand knob’ on the surface of M1 [1]. The anode is located over the C3 region on the scalp when attempting to increase excitability of the right-hand area of the M1 [2,3]. Moreover, the HD-tDCS stimulation protocol was obtained through computational simulation, and the simulation software could only provide standard electrode positions based on the 10-20 system. Regarding the distance between the hotspot and C3, we deeply apologize that this data was not completely recorded due to inadequate considerations during the research in 2021. According to existing literature, the distance between the hotspot location of FDI and C3 is generally less than 12.3±4.7mm [4]. To address this limitation, we have added the following content in the limitation section.
“6) Target areas differ. The locations of stimulation target areas differ between TI and HD-tDCS, which may be one of the reasons leading to discrepancies in research results.” (P16, line 493-495)
Reference:
[1] Yousry, T.A., Schmid, U.D., Alkadhi, H., Schmidt, D., Peraud, A., Buettner, A., Winkler, P., 1997. Localization of the motor hand area to a knob on the precentral gyrus. A new landmark. Brain 120, 141–157. https://doi.org/10.1093/brain/120.1.141
[2] Brunoni, A.R., Nitsche, M.A., Bolognini, N., Bikson, M., Wagner, T., Merabet, L., Edwards, D.J., Valero-Cabre, A., Rotenberg, A., Pascual-Leone, A., Ferrucci, R., Priori, A., Boggio, P.S., Fregni, F., 2012. Clinical research with transcranial direct current stimulation (tDCS): challenges and future directions. Brain stimul. 5, 175–195. https://doi.org/10.1016/j.brs.2011.03.002
[3] Silva, L.M., Silva, K.M.S., Lira-Bandeira, W.G., Costa-Ribeiro, A.C., Araújo-Neto, S.A., 2021. Localizing the primary motor cortex of the hand by the 10-5 and 10-20 systems for neurostimulation: an MRI study. Clin. EEG Neurosci. 52, 427–435. https://doi.org/10.1177/1550059420934590
[4] Kim, H., Wright, D. L., Rhee, J., & Kim, T. (2023). C3 in the 10-20 system may not be the best target for the motor hand area. Brain research, 1807, 148311.
Major comment 6. The references 12 and 13 are the same, and 13 does not relate to the statement concerning your expected effect of TI in the gamma band.
Respond to MC 6:
Thank you very much for your careful review and valuable suggestions. As you suggested, we have corrected the repetition of references 12 and 13 and have cited the correct reference in relation to the expected effect of TI in the gamma band. The correct reference is as follow.
“Furthermore, TI stimulation may result in frequency-specific modulations of neural oscillations, particularly in the gamma band, which has been associated with local circuit interactions and cognitive processing [1].” (P2, line 84-87)
Reference:
[1] Zheng, J.; Stevenson, R.F.; Mander, B.A.; Mnatsakanyan, L.; Hsu, F.P.K.; Vadera, S.; Knight, R.T.; Yassa, M.A.; Lin, J.J. Multiplexing of Theta and Alpha Rhythms in the Amygdala-Hippocampal Circuit Supports Pattern Separation of Emotional Information. Neuron 2019, 102, 887-898.e885, doi: 10.1016/j.neuron.2019.03.025.
Major comment 7. The age of your females and males are identical, this must be a typo.
Respond to MC 7:
Thank you for your careful review and valuable comments. We carefully checked the raw data of age, and indeed found a typo of the age information. We sincerely apologize for this oversight and have already made corrections to the relevant content in the text. Thank you again for your thorough examination of this study. The corrections are as follows:
“Subjects participated in our experiment, 9 females (age: 24.11 ± 0.93 years) and 31 males (age: 25.97 ± 3.53 years).” (P3, line 93-94)
Major comment 8. There is a lack of any specific references in your methods sections describing your tDCS and TI protocols respectively, please reference prior work.
Respond to MC 8:
Thank you for your valuable suggestion regarding the lack of specific references in our methods section. As you suggested, we added many specific references in your methods sections to support our tDCS and TI protocols. The added reference is as follows.
[1] Esmaeilpour Z, Shereen AD, Ghobadi-Azbari P, Datta A, Woods AJ, Ironside M, O'Shea J, Kirk U, Bikson M, Ekhtiari H. Methodology for tDCS integration with fMRI. Hum Brain Mapp. 2020 May;41(7):1950-1967. doi: 10.1002/hbm.24908. Epub 2019 Dec 24. PMID: 31872943; PMCID: PMC7267907.
[2] Conforto AB, Z'Graggen WJ, Kohl AS, Rösler KM, Kaelin-Lang A. Impact of coil position and electrophysiological monitoring on determination of motor thresholds to transcranial magnetic stimulation. Clin Neurophysiol. 2004 Apr;115(4):812-9. doi: 10.1016/j.clinph.2003.11.010. PMID: 15003761.
[3] Zhu, Z.; Xiong, Y.; Chen, Y.; Jiang, Y.; Qian, Z.; Lu, J.; Liu, Y.; Zhuang, J. Temporal Interference (TI) Stimulation Boosts Functional Connectivity in Human Motor Cortex: A Comparison Study with Transcranial Direct Current Stimulation (tDCS). Neural Plast 2022, 2022, 7605046, doi:10.1155/2022/7605046.

Reviewer 2 Report
Comments and Suggestions for Authors
The manuscript titled "Comparative Effects of Temporal Interference and High-Definition Transcranial Direct Current Stimulation on Spontaneous Neuronal Activity in the Primary Motor Cortex: A Randomized Crossover Study" presents a well-structured and methodologically rigorous investigation into the differential effects of Temporal Interference (TI) stimulation and High-Definition transcranial Direct Current Stimulation (HD-tDCS) on spontaneous neuronal activity. The study is novel in its approach, addressing an important gap in non-invasive brain stimulation research. By employing a randomized crossover design, the authors effectively minimize inter-subject variability, enhancing the reliability of their findings. The use of multiple neuroimaging markers (ReHo, dReHo, fALFF, dfALFF) provides a comprehensive assessment of neuronal activity, while statistical analyses, including false discovery rate correction, further strengthen the study’s credibility. The results are clearly presented, with TI demonstrating more significant and sustained effects on spontaneous neuronal activity compared to HD-tDCS. These findings have potential implications for optimizing neuromodulation techniques for therapeutic applications.
Despite these strengths, the study has several limitations:
- One major issue is the absence of behavioral data, which weakens its real-world applicability. Although the neurophysiological changes are well-documented, the lack of cognitive or motor function assessments makes it difficult to determine the practical implications of the findings.
- Additionally, the small and homogeneous sample (40 right-handed participants, with a gender imbalance of 31 males and 9 females) limits the generalizability of the results..
- Another critical limitation is the lack of a sham control condition. Without a placebo group, it is difficult to rule out non-specific effects or baseline fluctuations in brain activity, which could have influenced the observed differences between TI and HD-tDCS.
- Furthermore, the manuscript does not explicitly address whether the stimulation sessions were counterbalanced to avoid order effects, which is essential in crossover designs to prevent carryover influences.
- The exclusion of eight participants due to excessive head motion in fMRI further raises concerns about potential selection bias.
- The discussion section could benefit from a stronger mechanistic explanation of the differences between TI and HD-tDCS. While the study suggests that TI exerts a more potent and lasting effect due to its deeper cortical penetration, a more detailed exploration of the underlying neurophysiological mechanisms would be valuable.
- Computational modeling or electric field distribution analyses could further elucidate why TI produces distinct modulation patterns.
- Additionally, the manuscript contains minor grammatical inconsistencies, redundant phrasing, and formatting issues in some figures and tables. The writing could be streamlined to improve clarity, and figures should include more detailed legends to aid reader comprehension.
- The study does not provide sufficient details on the stimulation parameters, such as electrode positioning and current density variations, which are crucial for reproducibility. More transparency in reporting these methodological aspects would enhance the study’s credibility.
- The paper lacks a thorough discussion on potential confounding variables, such as individual differences in skull thickness, baseline cortical excitability, and prior exposure to non-invasive brain stimulation techniques, which may have influenced the results.
- While the study focuses on spontaneous neuronal activity changes, it does not explore whether these alterations persist beyond the stimulation period. A follow-up investigation on the duration of effects would provide more insight into the long-term impact of TI and HD-tDCS.
- There is limited discussion on the safety and tolerability of TI compared to HD-tDCS. Given the increasing interest in TI as a neuromodulation technique, more information on participant-reported adverse effects would be valuable.
- The statistical power of the study is not explicitly addressed. Given the small sample size, an analysis of whether the study was adequately powered to detect meaningful differences between conditions would be beneficial.
- The study does not compare its findings with other existing neuromodulation techniques, such as transcranial magnetic stimulation (TMS). A broader contextualization of results would help situate the study within the field of non-invasive brain stimulation.
- The authors do not discuss the translational potential of TI and HD-tDCS for clinical applications, such as stroke rehabilitation or neuropsychiatric disorders. Including this perspective would increase the manuscript’s relevance for a wider audience.
Comments on the Quality of English Language
The manuscript contains minor grammatical inconsistencies, redundant phrasing, and formatting issues in some figures and tables. The writing could be streamlined to improve clarity, and figures should include more detailed legends to aid reader comprehension.
Author Response
Response to Reviewer 2 Comments
We would like to sincerely thank the reviewers for their helpful recommendations. We have seriously considered all the comments and carefully revised the manuscript accordingly. All revisions are marked using track changes in the revised PDF document for review. In addition, we have enhanced the language of the manuscripts. We feel that the quality of the manuscript has been significantly improved with these modifications and improvements based on the reviewers’ suggestions and comments. We hope our revision will lead to an acceptance of our manuscript for publication in the Brain Science.
General comment
The manuscript titled "Comparative Effects of Temporal Interference and High-Definition Transcranial Direct Current Stimulation on Spontaneous Neuronal Activity in the Primary Motor Cortex: A Randomized Crossover Study" presents a well-structured and methodologically rigorous investigation into the differential effects of Temporal Interference (TI) stimulation and High-Definition transcranial Direct Current Stimulation (HD-tDCS) on spontaneous neuronal activity. The study is novel in its approach, addressing an important gap in non-invasive brain stimulation research. By employing a randomized crossover design, the authors effectively minimize inter-subject variability, enhancing the reliability of their findings. The use of multiple neuroimaging markers (ReHo, dReHo, fALFF, dfALFF) provides a comprehensive assessment of neuronal activity, while statistical analyses, including false discovery rate correction, further strengthen the study’s credibility. The results are clearly presented, with TI demonstrating more significant and sustained effects on spontaneous neuronal activity compared to HD-tDCS. These findings have potential implications for optimizing neuromodulation techniques for therapeutic applications.
Respond to GC 1:
We are deeply grateful for your affirmation of our work and for providing us with many professional review comments. We have revised the paper according to each of your professional review comments. With your help, the quality of our paper has significantly improved.
Major comment 1. One major issue is the absence of behavioral data, which weakens its real-world applicability. Although the neurophysiological changes are well-documented, the lack of cognitive or motor function assessments makes it difficult to determine the practical implications of the findings.
Respond to MC 1:
Thank you very much for your valuable suggestion. We fully agree with your comments regarding the absence of behavioral data as a limitation in our study. In response, we have acknowledged this limitation in the revised manuscript's "Limitations" section, emphasizing that the lack of cognitive and motor function assessments restricts the direct translation of neurophysiological findings to real-world applications. We further propose in the "Future Directions" section that subsequent studies could systematically integrate behavioral metrics (e.g., cognitive testing batteries and motor performance evaluations) to cover this critical gap. The added content is as follows.
“1) Without collecting behavioral data. The study lacks assessments of cognitive and motor function, which restricts the direct translation of neurophysiological findings to real-world applications. Future studies could systematically integrate behavioral metrics (e.g., cognitive testing batteries and motor performance evaluations) to address this limitation.” (P15, line 474-478)
Major comment 2. Additionally, the small and homogeneous sample (40 right-handed participants, with a gender imbalance of 31 males and 9 females) limits the generalizability of the results.
Respond to MC 2:
We appreciate your insightful comment regarding the sample characteristics. We fully agree that the limited sample size and homogeneity (40 right-handed participants with 31 males and 9 females) may affect the generalizability of findings. We have addressed this limitation in the limitation section. We revised our manuscript as follow.
- The limitation of the sample characteristics:
“2) Sample characteristics, the small sample size and relatively uneven gender distribution may limit the generalizability of our findings to broader populations.” (P15, line 479-483)
- The detail of sample calculation:
To demonstrate the reliability of our sample size, we have added details regarding the sample size calculation in the Methods section.
“The sample size was determined through a priori power analysis using G*Power 3.1 software [1], to ensure sufficient statistical power for testing the primary hypothesis. The calculation was based on an effect size of f = 0.25, a significance level of α= 0.05, and a power of (1 -β) = 0.80. The required sample size is 34, and with an additional 20% dropout rate, the final sample size is 40” (P3, line 94-98)
Reference:
- Faul, F., Erdfelder, E., Lang, A.-G., & Buchner, A. (2007). G*Power 3: A flexible statistical power analysis program for the social, behavioral, and biomedical sciences. Behavior Research Methods, 39(2), 175-191. https://doi.org/10.3758/BF03193146
Major comment 3. Another critical limitation is the lack of sham control condition. Without a placebo group, it is difficult to rule out non-specific effects or baseline fluctuations in brain activity, which could have influenced the observed differences between TI and HD-tDCS.
Respond to MC 3:
Thank you for your valuable suggestion. We acknowledge and agree that the absence of a sham stimulation condition is a significant limitation in our study. In response to your critical comments, we have incorporated limitation in the revised manuscript. The added content is as follow:
“4) lack of sham condition and proprioception data. Without sham condition, the potential confounding influence of scalp sensation associated with active stimulation cannot be excluded and the experimental design did not isolate the contribution of high-frequency carrier signals from the intended low-frequency envelope modulation. To address both limitations, future studies should implement a control sham condition: The sham condition using two identical high-frequency carriers (e.g., 2000 Hz) to exclude the effects of high-frequency stimulation.” (P16, line 483-489)
Major comment 4. Furthermore, the manuscript does not explicitly address whether the stimulation sessions were counterbalanced to avoid order effects, which is essential in crossover designs to prevent carryover influences.
Respond to MC 4:
We sincerely appreciate your careful review. In our study, we implemented a randomized crossover design with counterbalanced stimulation sequences to mitigate order effects. This was not explicitly detailed in the original manuscript. In the revised version, we have added the detail of counterbalanced.
“To prevent order effects, the participants were randomly assigned to receive the two different types of stimulation by an experimenter who not involved in data processing.” (P3, line 117-119)
Major comment 5. The exclusion of eight participants due to excessive head motion in fMRI further raises concerns about potential selection bias.
Respond to MC 5:
Thank you for your comments regarding the potential selection bias due to the exclusion of eight participants with excessive head motion in our fMRI study. In our initial analysis, we adopted a more stringent threshold (2 mm translation and 2° rotation) for head motion to ensure the quality of the data, given that head motion can have a significant impact on fMRI results. To address this concern, we have re-evaluated our data using a more relaxed threshold (3 mm translation and 3° rotation). The re-evaluated results show no difference from the original results in activation regions, with only a slight variation in the number of activated voxels. The re-evaluated results is as follow.
“S4. Statistical results under different head movement exclusion criteria
Three participants were excluded from the study due to excessive head motion, with exclusion criteria set at a maximum displacement of over 3.0 mm or an angular rotation exceeding 3.0°.
Supplemental Table1. Significant differences in ReHo. (N=37)
|
Comparisons |
Brain regions/BA |
Peak MNI coordinates |
Cluster Voxels |
Peak t values |
||
|
|
|
X |
y |
z |
|
|
|
tDCS S3 - tDCS S1 |
Precentral_R |
39 |
-27 |
60 |
55 |
4.80 |
|
|
Temporal_Sup_R |
54 |
-9 |
0 |
22 |
4.38 |
|
|
Postcentral_L |
-27 |
-39 |
66 |
94 |
4.33 |
|
|
Supp_Motor_Area_R |
6 |
-18 |
60 |
28 |
4.51 |
|
|
Postcentral_R |
30 |
-39 |
69 |
42 |
3.92 |
|
tDCS S3 - tDCS S4 |
Postcentral_R |
45 |
-27 |
60 |
18 |
4.66 |
|
|
Temporal_Sup_R |
54 |
-12 |
3 |
29 |
4.62 |
|
TI S2 - TI S1 |
Precentral_L |
-33 |
-30 |
60 |
79 |
5.30 |
|
|
Postcentral_L |
-51 |
-18 |
39 |
24 |
4.34 |
|
|
Insula_R |
48 |
-3 |
0 |
27 |
4.10 |
|
|
Insula_R |
36 |
-15 |
15 |
20 |
3.92 |
|
TI S3 - TI S1 |
Temporal_Sup_L |
-45 |
-24 |
9 |
181 |
5.44 |
|
|
Postcentral_R |
30 |
-33 |
60 |
284 |
6.30 |
|
|
Postcentral_L |
-27 |
-33 |
60 |
378 |
5.12 |
|
|
BA48_R |
33 |
-18 |
12 |
192 |
5.00 |
|
|
Supp_Motor_Area_R |
6 |
-15 |
54 |
20 |
4.47 |
|
TI S4 - TI S1 |
Precentral_R |
51 |
-18 |
45 |
92 |
5.13 |
|
|
Postcentral_L |
-48 |
-18 |
51 |
52 |
4.65 |
|
TI S3 - TI S2 |
Temporal_Sup_L |
-48 |
-15 |
0 |
46 |
5.00 |
|
|
Postcentral_R |
30 |
-33 |
60 |
38 |
4.71 |
|
|
Postcentral_L |
-33 |
-42 |
57 |
22 |
4.29 |
|
TI S3 - TI S4 |
Postcentral_R |
30 |
-33 |
60 |
42 |
5.14 |
|
|
Temporal_Sup_R |
60 |
-33 |
9 |
25 |
5.00 |
|
|
BA48_R |
33 |
-18 |
9 |
61 |
4.78 |
|
|
Heschl_L |
-45 |
-15 |
6 |
67 |
4.41 |
Notes. BA, Brodmann’s area, L, left; R, right; S1: baseline; S2: first half of the stimulus; S3: second half of the stimulus; S4: post-stimulus; TI, Temporal Interference stimulation; tDCS, transcranial Direct Current Stimulation; ReHo, Regional Homogeneity; Interaction, the stimulation type× time interaction effect assessed via two-way repeated measures ANOVA. Report results with edge - level p<0.001 and cluster - level p<0.05.
Supplemental Figure 4. The significant differences in brain regions of ReHo. (a) Between tDCS-S3 and tDCS-S1. (b) Between tDCS-S3 and tDCS-S4. (c) Between TI-S3 and TI-S1. (d) Between TI-S3 and TI-S2. (e) Between TI-S4 and TI-S1. (f) Between TI-S3 and TI-S2. (g)Between TI-S3 and TI-S4 Note: The color bar represents the T value; S1: baseline; S2: first half of the stimulus; S3: second half of the stimulus; S4: post-stimulus; TI, Temporal Interference stimulation; tDCS, transcranial Direct Current Stimulation; A, anterior; R, right; P, posterior. ReHo, Regional Homogeneity; Report results with edge - level p<0.001 and cluster - level p<0.05.
Supplemental Table2. Significant differences in dReHo.
|
Comparisons |
Brain regions/BA |
Peak MNI coordinates |
Cluster Voxels |
Peak t values |
||
|
X |
y |
z |
||||
|
tDCS S1 – tDCS S3 |
Temporal_Sup_R |
54 |
-18 |
6 |
29 |
4.90 |
|
|
Postcentral_L |
-33 |
-45 |
66 |
29 |
4.64 |
|
|
Supp_Motor_Area_R |
6 |
-21 |
57 |
38 |
4.51 |
|
|
Temporal_Sup_L |
-57 |
-21 |
6 |
37 |
4.50 |
|
|
Precentral_R |
48 |
-15 |
39 |
34 |
4.43 |
|
tDCS S4 – tDCS S3 |
Precentral_R |
24 |
-30 |
72 |
188 |
5.18 |
|
|
Temporal_Sup_R |
57 |
-9 |
3 |
54 |
4.48 |
|
TI S1 – TI S3 |
Heschl_L |
-42 |
-27 |
9 |
51 |
5.52 |
|
|
Precuneus_L |
-15 |
-45 |
66 |
45 |
5.18 |
|
|
Postcentral_R |
27 |
-30 |
60 |
68 |
5.07 |
|
|
Postcentral_L |
-45 |
-18 |
51 |
69 |
4.98 |
|
|
Paracentral_Lobule_L |
-6 |
-36 |
72 |
39 |
4.85 |
|
|
Postcentral_R |
12 |
-39 |
72 |
24 |
4.78 |
|
|
Heschl_R |
48 |
-18 |
9 |
67 |
4.60 |
|
TI S4 – TI S3 |
Temporal_Sup_R |
60 |
-36 |
9 |
21 |
5.03 |
|
|
Heschl_R |
48 |
-18 |
9 |
29 |
4.74 |
|
|
Supp_Motor_Area_R |
6 |
-3 |
45 |
23 |
4.49 |
|
|
Temporal_Sup_L |
-42 |
-24 |
6 |
50 |
4.45 |
|
|
Postcentral_R |
27 |
-27 |
57 |
25 |
4.43 |
Notes. BA, Brodmann’s area, L, left; R, right; S1: baseline; S2: first half of the stimulus; S3: second half of the stimulus; S4: post-stimulus;; TI, Temporal Interference stimulation; tDCS, transcranial Direct Current Stimulation. dReHo, dynamic, Regional Homogeneity; Interaction, the stimulation type× time interaction effect assessed via two-way repeated measures ANOVA. Report results with edge - level p<0.001 and cluster - level p<0.05.
Supplemental Figure 5. The significant differences in brain regions of dReHo. (a) Between tDCS-S1 and tDCS-S3. (b) Between tDCS-S4 and tDCS-S3. (c) Between TI-S1 and TI-S3. (d) Between TI-S4 and TI-S3. Note: The color bar represents the T value. S1: baseline; S3: second half of the stimulus; S4: post-stimulus; dReHo, dynamic ReHo; TI, Temporal Interference stimulation; tDCS, transcranial Direct Current Stimulation; A, anterior; R, right; P, posterior; S, superior. Report results with edge - level p<0.001 and cluster - level p<0.05.”
Major comment 6. The discussion section could benefit from a stronger mechanistic explanation of the differences between TI and HD-tDCS. While the study suggests that TI exerts a more potent and lasting effect due to its deeper cortical penetration, a more detailed exploration of the underlying neurophysiological mechanisms would be valuable.
Respond to MC 6:
Thank you for this insightful suggestion that strengthens the mechanistic interpretation of our findings. We fully agree that elucidating the neurophysiological distinctions between TI and HD-tDCS is crucial. In the revised Discussion section, we have added:
“In the comparison of tES, tDCS employs 1-2 mA weak direct current to directionally modulate brain region activity: the anode enhances neuronal excitability to promote regional functionality, while the cathode inhibits neuronal activity, achieving sustained improvements in neural circuit excitability and motor function [1]. tACS utilizes sinusoidal electric fields with periodic oscillations to forcibly reorganize the firing rhythms of neuronal populations, promoting phase synchronization between neuronal discharges and external stimuli. By applying 20 Hz alternating current synchronized with the brain's beta rhythm (13-30 Hz), it enables targeted modulation of intrinsic neural oscillation patterns in motor control-related brain regions (e.g., the motor cortex), optimizing motor function [2]. However, both stimulation modalities show limited electric field focality. To overcome the limitations of focality, TI stimulation utilizes two slightly frequency different high-frequency currents (2000Hz and 2020Hz) to generate a low-frequency (20Hz) envelope that can penetrate deep brain regions and achieve a more focal stimulation. The same to tACS, TI modulate brain intrinsic neural oscillation to enhance motor function [3].” (P15, line 444-457)
Reference:
- Pelletier, S. J., & Cicchetti, F. (2014). Cellular and molecular mechanisms of action of transcranial direct current stimulation: evidence from in vitro and in vivo models. The international journal of neuropsychopharmacology, 18(2), pyu047. https://doi.org/10.1093/ijnp/pyu047
- Ma, R., Xia, X., Zhang, W., Lu, Z., Wu, Q., Cui, J., Song, H., Fan, C., Chen, X., Zha, R., Wei, J., Ji, G. J., Wang, X., Qiu, B., & Zhang, X. (2022). High Gamma and Beta Temporal Interference Stimulation in the Human Motor Cortex Improves Motor Functions. Frontiers in neuroscience, 15, 800436. https://doi.org/10.3389/fnins.2021.800436
- Zhu, Z., & Yin, L. (2023). A mini-review: recent advancements in temporal interference stimulation in modulating brain function and behavior. Frontiers in human neuroscience, 17, 1266753. https://doi.org/10.3389/fnhum.2023.1266753
Major comment 7. Computational modeling or electric field distribution analyses could further elucidate why TI produces distinct modulation patterns.
Respond to MC 7:
We sincerely appreciate your insightful suggestion regarding the TI exposure simulations. As you suggested, we used finite element modeling software (COMSOL) to simulate the TI exposure and HD-tDCS exposure, measuring the field strength within a spherical region of 10 mm radius centered at the precentral gyrus (-42, -13, 53). The results revealed that the envelope field intensity for TI was 0.541 V/m, while the field intensity for HD-tDCS was 0.344 V/m (Figure 4). We have added the simulation methodology and result in the supplementary material as follows.
S2. Computational simulation of electrical field intensity
To elucidate the spatial electric field distribution in the human brain during transcranial electrical stimulation (TIS and tDCS), we employed finite element modeling software (COMSOL) to simulate electric field propagation. An anatomically derived head model based on the Colin27 SimNIBS template was utilized. The computational model was stratified into five discrete tissue compartments with distinct electrical conductivities: scalp (0.333 S/m), skull (0.008 S/m), cerebrospinal fluid (CSF) (1.79 S/m), gray matter (0.4 S/m), and white matter (0.15 S/m). Electrode configurations for both stimulation protocols were illustrated in Supplemental Figures 3 and 4. The spatial distribution of amplitude-modulated electric fields was quantified along the posterior-anterior axis. A quasi-static approximation of Maxwell's equations was assumed in the simulation (Huang, Y., & Parra, L. C. 2019).
“The simulated current intensity of TI and HD-tDCS was measured within a spherical region with a 10-mm radius centered at the precentral gyrus coordinates (-42, -13, 53). The results revealed that the envelope field intensity for TI was 0.541 V/m (Figure 2C), whereas the field intensity for HD-tDCS was 0.344 V/m (Figure 3C).” (P9, L27)
Supplemental Figure 3. Simulation model, electrode placement, and electrical field. (A) Head model with electrode placements. (B) Electrode locations based on 10-20 system. (C) Electric field simulation diagram of TI. Note: Stimulation Pair 1 A1-A2 (blue, 2000 Hz channel, 2 mA), Pair 2 B1-B2 (red, 2020 Hz channel, 2 mA). Grey electrode is the targeting area.
Supplemental Figure 4. Simulation model, electrode placement, and electrical field. (A) Head model with cathode (blue) and anode (red) electrodes. (B) Electrode locations based on 10-20 system: Anode (C3, 2000 μA), Cathodes (T7, -684 μA; P3, -774 μA; Cz, -542 μA). C) Electric field simulation diagram of HD-tDCS.
Major comment 8. Additionally, the manuscript contains minor grammatical inconsistencies, redundant phrasing, and formatting issues in some figures and tables. The writing could be streamlined to improve clarity, and figures should include more detailed legends to aid reader comprehension.
Respond to MC 8:
We sincerely appreciate your detailed review of our manuscript. As you suggested, we have checked the grammar by the “Paperpal” throughout the document to maintain consistency. We have also reviewed the language for clarity and conciseness to ensure clear expression. Additionally, we have corrected the format of some images and added more detailed captions to the figures to help readers better understand the content.
Major comment 9. The study does not provide sufficient details on the stimulation parameters, such as electrode positioning and current density variations, which are crucial for reproducibility. More transparency in reporting these methodological aspects would enhance the study’s credibility.
Respond to MC 9:
Thank you for your valuable comments. As you suggested, we have added the detailed information on the stimulation parameters in the supplementary files. These supplementary materials provide comprehensive data on the stimulation paradigms used in our study, ensuring that all methodological aspects are thoroughly documented. We believe that this additional information will enhance the clarity and credibility of our research. The added information is as follows.
S3. Details of the stimulus paradigm
Temporal Interference Stimulation:
Temporal Interference Stimulation involves several key steps. First, Transcranial Magnetic Stimulation (TMS) is used for target point identification. A TMS device, set to the lowest possible stimulation intensity for safety and comfort, targets the primary motor cortex left. The target point is determined by stimulating different regions of this cortex and identifying the location that induces a visible contraction in the first dorsal interosseous muscle of the right hand, confirmed by visual inspection and EMG recordings if needed [1]. Electrode placement for Transcranial Impedance (TI) stimulation is carefully planned. Four electrodes form a 4 - cm - sided square around the target point identified through TMS. The stimulation was administered via four MRI-compatible conductive rubber electrodes, each measuring 1.5 cm × 2 cm. The square's orientation has two sides parallel to the line from the glabella to the inion and the other two parallel to the line connecting the ears, ensuring consistency and reproducibility. Electrodes are securely attached to the scalp with conductive gel and adhesive to ensure good electrical contact, and the impedance between each pair is measured and recorded before stimulation. Then, the stimulation parameters for TI stimulation are set. The R1-R2 channel is set to 2000 Hz and the L1-L2 channel to 2020 Hz, based on previous research and study requirements. The envelope wave generated by these two channels has a difference frequency of 20 Hz, with an amplitude of 2 mA in each channel, modulating neural activity in the target region [2]. A Soterix Medical device from New Jersey, USA, is used for its precision and safety features. The stimulation session protocol is established. The session lasts 20 minutes, determined by the desired effect and participant tolerability. It includes two brief 30 - second ramp-up and ramp-down phases at the beginning and end. The ramp-up phase gradually increases stimulation intensity to the target level, while the ramp-down phase gradually decreases it to zero, minimizing discomfort and adverse effects and ensuring a smooth transition into and out of stimulation.
High-Definition Transcranial Direct Current Stimulation:
The experimental protocol used an MRI-compatible DC-STIMULATOR PLUS (NeuroConn GmbH, Ilmenau, Germany) for HD-tDCS. The device settings were configured based on the parameters outlined in Esmaeilpour et al. [3]. The stimulation was administered via four MRI-compatible conductive rubber electrodes, each measuring 1.5 cm × 2 cm. Electrode placement was performed using an international 10-20 EEG system for brain localization. The STIMWEAR software simulated the stimulation montage with the left FDI-M1 as the target region, four distributed electrodes, and a total current of 2 mA. The stimulation montage obtained was as follows: 2000 μA at C3 (anode), -774 μA at P3, -684 μA at T7, and -542 μA at Cz (cathode). The total stimulation time was 20 min, which included a 1-minute ramp-up and ramp-down phase (30 s each), with real-time impedance maintained below 30 kΩ.
Reference:
- Conforto, A.B.; Z'Graggen, W.J.; Kohl, A.S.; Rösler, K.M.; Kaelin-Lang, A. Impact of coil position and electrophysiological monitoring on determination of motor thresholds to transcranial magnetic stimulation. Clin Neurophysiol 2004, 115, 812-819, doi:10.1016/j.clinph.2003.11.010.
- Violante, I.R.; Alania, K.; Cassarà, A.M.; Neufeld, E.; Acerbo, E.; Carron, R.; Williamson, A.; Kurtin, D.L.; Rhodes, E.; Hampshire, A.; et al. Non-invasive temporal interference electrical stimulation of the human hippocampus. Nat Neurosci 2023, 26, 1994-2004, doi:10.1038/s41593-023-01456-8.
- Esmaeilpour, Z.; Shereen, A.D.; Ghobadi-Azbari, P.; Datta, A.; Woods, A.J.; Ironside, M.; O'Shea, J.; Kirk, U.; Bikson, M.; Ekhtiari, H. Methodology for tDCS integration with fMRI. Hum Brain Mapp 2020, 41, 1950-1967, doi:10.1002/hbm.24908.
Major comment 10. The paper lacks a thorough discussion on potential confounding variables, such as individual differences in skull thickness, baseline cortical excitability, and prior exposure to non-invasive brain stimulation techniques, which may have influenced the results.
Respond to MC 10:
Thank you very much for your professional and detailed review comments. We fully agree that differences in skull thickness, cortical excitability, and prior exposure to non-invasive brain stimulation techniques could influence the study results. Therefore, we have required that participants avoid strenuous exercise on the day before testing, refrain from consuming beverages or medications that may alter neural excitability (such as coffee and alcohol) within 4 hours prior to testing and ensure an interval of at least 48 hours between stimulations to wash-out any stimulation effects. Additionally, we recognize that individual differences may have a significant impact on the results. Thus, in the limitations section, we emphasized the importance of adopting personalized stimulation protocols. The added content is as follows:
“3) Individual differences. Individual differences, such as skull thickness, may significantly influence the stimulating effects of TI. Future studies should investigate personalized stimulation protocols to optimize the efficacy of TI.” (P15, line 480-483)
Major comment 11. While the study focuses on spontaneous neuronal activity changes, it does not explore whether these alterations persist beyond the stimulation period. A follow-up investigation on the duration of effects would provide more insight into the long-term impact of TI and HD-tDCS.
Respond to MC 11:
Thank you for your insightful suggestion. We fully agree with the importance of investigating the post effects of stimulation. And we have highlighted the importance of examining long term effects in the limitations section of our revised manuscript. The added content is as follows:
“5) Without investigating the long-term effect. The current study did not investigate whether the observed changes in spontaneous neuronal activity persisted beyond the stimulation period. Future studies should incorporate longer follow-up periods and repeated measurements to determine the duration and stability of these effects. This would provide valuable insights into the potential long-term impact of TI and HD-tDCS on neuronal activity.” (P16, line 489-493)
Major comment 12. There is limited discussion on the safety and tolerability of TI compared to HD-tDCS. Given the increasing interest in TI as a neuromodulation technique, more information on participant-reported adverse effects would be valuable.
Respond to MC 12:
Thank you for raising this important concern regarding the limited discussion on the safety and tolerability of TI compared to HD-tDCS. We have conducted additional experiments to address this issue. The detailed information on the participant-reported adverse effects has been included in the supplementary file.
S1. Blind effect and security testing
Fifteen healthy adults (male, aged 20.3 ± 1.1 years) were recruited using a double-blind crossover design. Each participant randomly received two types of stimulation: TIS ((2000 Hz, 2020 Hz, frequency difference 20 Hz), 2 mA per pair, total current: 4 mA) and HD-tDCS (total current: 2 mA). Each stimulation session lasted 6 minutes (including 30 seconds of ramp-up and ramp-down), with a 48-hour washout period between sessions. After stimulation, participants completed the subject subjective assessment scale and blinding test questionnaire. Using Pearson's chi-square test to analyze the questionnaire results, the subject subjective assessment scale revealed no significant difference in total adverse reaction scores between the two groups (χ2 = 11.121, P = 0.113) (Supplemental Figure 1). Similarly, the blinding test questionnaire showed no significant difference in stimulation blinding across groups (χ2 = 6.00, P = 0.107) (Supplemental Figure 2).
Supplemental Figure 1. Adverse reaction scores comparison between TIS group and HD-tDCS groups. Note: TIS, temporal interference stimulation; HD-tDCS, high-definition transcranial direct current stimulation.
Supplemental Figure 2. Blinding efficacy assessment between TIS group and HD-tDCS groups. Note: TIS, temporal interference stimulation; HD-tDCS, high-definition transcranial direct current stimulation.
Major comment 13. The statistical power of the study is not explicitly addressed. Given the small sample size, an analysis of whether the study was adequately powered to detect meaningful differences between conditions would be beneficial.
Respond to MC 13:
Thank you for highlighting this crucial methodological consideration. We apologize for not clearly detailing the sample size calculation in the Methods section. The additional comparative content is as follows.
“Sample size was determined through a priori power analysis using G*Power 3.1 software [1], to ensure sufficient statistical power for testing the primary hypothesis. The calculation was based on an effect size of f = 0.25, a significance level of α= 0.05, and a power of (1 -β) = 0.80. The required sample size is 34, and with an additional 20% dropout rate, the final sample size is 40.” (P3, line 94-98)
Reference:
- Faul, F., Erdfelder, E., Lang, A.-G., & Buchner, A. (2007). G*Power 3: A flexible statistical power analysis program for the social, behavioral, and biomedical sciences. Behavior Research Methods, 39(2), 175-191. https://doi.org/10.3758/BF03193146
Major comment 14. The study does not compare its findings with other existing neuromodulation techniques, such as transcranial magnetic stimulation (TMS). A broader contextualization of results would help situate the study within the field of non-invasive brain stimulation.
Respond to MC 14:
Thank you very much for your suggestions. As you suggested, we have compared the results of this study with other existing neuromodulation techniques. The additional comparative content is as follows.
“In the comparison of tES, tDCS employs 1-2 mA weak direct current to directionally modulate brain region activity: the anode enhances neuronal excitability to promote regional functionality, while the cathode inhibits neuronal activity, achieving sustained improvements in neural circuit excitability and motor function [1]. tACS utilizes sinusoidal electric fields with periodic oscillations to forcibly reorganize the firing rhythms of neuronal populations, promoting phase synchronization between neuronal discharges and external stimuli. By applying 20 Hz alternating current synchronized with the brain's beta rhythm (13-30 Hz), it enables targeted modulation of intrinsic neural oscillation patterns in motor control-related brain regions (e.g., the motor cortex), optimizing motor function [2]. However, both stimulation modalities show limited electric field focality. To overcome the limitations of focality, TI stimulation utilizes two slightly frequency different high-frequency currents (2000Hz and 2020Hz) to generate a low-frequency (20Hz) envelope that can penetrate deep brain regions and achieve a more focal stimulation. The same to tACS, TI modulate brain intrinsic neural oscillation to enhance motor function [3].” (P15, line 444-457)
Reference:
- Pelletier, S. J., & Cicchetti, F. (2014). Cellular and molecular mechanisms of action of transcranial direct current stimulation: evidence from in vitro and in vivo models. The international journal of neuropsychopharmacology, 18(2), pyu047. https://doi.org/10.1093/ijnp/pyu047
- Ma, R., Xia, X., Zhang, W., Lu, Z., Wu, Q., Cui, J., Song, H., Fan, C., Chen, X., Zha, R., Wei, J., Ji, G. J., Wang, X., Qiu, B., & Zhang, X. (2022). High Gamma and Beta Temporal Interference Stimulation in the Human Motor Cortex Improves Motor Functions. Frontiers in neuroscience, 15, 800436. https://doi.org/10.3389/fnins.2021.800436
- Zhu, Z., & Yin, L. (2023). A mini-review: recent advancements in temporal interference stimulation in modulating brain function and behavior. Frontiers in human neuroscience, 17, 1266753. https://doi.org/10.3389/fnhum.2023.1266753
Major comment 15. The authors do not discuss the translational potential of TI and HD-tDCS for clinical applications, such as stroke rehabilitation or neuropsychiatric disorders. Including this perspective would increase the manuscript’s relevance for a wider audience.
Respond to MC 15:
Thank you for your valuable suggestion. We fully agree that the addition of the translational medicine perspective will help enhance the clinical relevance of the research. In the discussion section of the revised manuscript, we have added the following content.
“The findings highlight the translational potential of TI and HD-tDCS for clinical applications. Both techniques demonstrated significant modulation of brain functional connectivity and spontaneous activity, essential for improving motor and cognitive functions [1]. In stroke rehabilitation, TI's stronger and more sustained effects on regions like the postcentral and superior temporal gyri suggest its potential to enhance neuroplasticity in sensorimotor networks. This could accelerate recovery of motor functions and sensory integration [2]. Similarly, HD-tDCS’s modulation of cortical excitability may complement traditional therapies by reinforcing neural circuits responsible for motor control [3]. For neuropsychiatric disorders, TI and HD-tDCS showed the ability to alter low-frequency oscillations and strengthen local synchrony, mechanisms linked to mood regulation and cognitive improvement [4]. TI’s superior focality and deeper penetration might allow better targeting of dysfunctional brain areas, such as the basal ganglia, implicated in Parkinson's disease and Huntington's disease [5]” (P15, line 460-472)
Reference:
[1] Roy, A., Baxter, B., & He, B. (2014). High-definition transcranial direct current stimulation induces both acute and persistent changes in broadband cortical synchronization: a simultaneous tDCS-EEG study. IEEE transactions on bio-medical engineering, 61(7), 1967–1978. https://doi.org/10.1109/TBME.2014.2311071
[2] Wu, C. W., Lin, S. N., Hsu, L. M., Yeh, S. C., Guu, S. F., Lee, S. H., & Chen, C. C. (2020). Synchrony Between Default-Mode and Sensorimotor Networks Facilitates Motor Function in Stroke Rehabilitation: A Pilot fMRI Study. Frontiers in neuroscience, 14, 548. https://doi.org/10.3389/fnins.2020.00548
[3] Morya, E., Monte-Silva, K., Bikson, M., Esmaeilpour, Z., Biazoli, C. E., Jr, Fonseca, A., Bocci, T., Farzan, F., Chatterjee, R., Hausdorff, J. M., da Silva Machado, D. G., Brunoni, A. R., Mezger, E., Moscaleski, L. A., Pegado, R., Sato, J. R., Caetano, M. S., Sá, K. N., Tanaka, C., Li, L. M., … Okano, A. H. (2019). Beyond the target area: an integrative view of tDCS-induced motor cortex modulation in patients and athletes. Journal of neuroengineering and rehabilitation, 16(1), 141. https://doi.org/10.1186/s12984-019-0581-1
[4] Alonzo, A., Brassil, J., Taylor, J. L., Martin, D., & Loo, C. K. (2012). Daily transcranial direct current stimulation (tDCS) leads to greater increases in cortical excitability than second daily transcranial direct current stimulation. Brain stimulation, 5(3), 208–213. https://doi.org/10.1016/j.brs.2011.04.006
[5] Stefania, E., Sirius B., Chris P., Romina B., Jill F., Mark J., Stuart C., Kinan M., Campbell L., Richard A., Johannes C., Masud H., Andrea H.h, Michele T., Gwenaëlle D. (2023). Subthalamic nucleus shows opposite functional connectivity pattern in Huntington’s and Parkinson’s disease, Brain Communications, 5(6), fcad282, https://doi.org/10.1093/braincomms/fcad282

Reviewer 3 Report
Comments and Suggestions for Authors
This study aims to compare the effects of High-Definition transcranial Direct Current Stimulation (HD-tDCS) and Temporal Interference (TI) stimulation on spontaneous neuronal activity in the primary motor cortex (M1). The Authors hypothesize that these two stimulation methods will produce distinct patterns of modulation in spontaneous neuronal activity within the M1 region. Assessing neural activity is important for evaluating information processing, neuroplasticity, and cognitive performance in different health states and pathological conditions.
Participants underwent two 20-minute sessions of TI or tDCS stimulation. The order in which the different stimulation types were administered must be specified or whether they were randomized. Neuronal activity was assessed using both functional and structural imaging methods.
The article is written clearly, has a clear structure. The conclusions correspond to the results. There are minor grammatical errors. A grammar check is necessary. Pay attention to sentence structure and spelling. You should also check for abbreviations.
Comments on the Quality of English LanguageA grammar check is necessary. Pay attention to sentence structure and spelling. You should also check for abbreviations.
Author Response
Response to Reviewer 3 Comments
We would like to sincerely thank the reviewers for their helpful recommendations. We have seriously considered all the comments and carefully revised the manuscript accordingly. All revisions are marked using track changes in the revised PDF document for review. In addition, we have enhanced the language of the manuscripts. We feel that the quality of the manuscript has been significantly improved with these modifications and improvements based on the reviewers’ suggestions and comments. We hope our revision will lead to an acceptance of our manuscript for publication in the Brain Science.
General comment
This study aims to compare the effects of High-Definition transcranial Direct Current Stimulation (HD-tDCS) and Temporal Interference (TI) stimulation on spontaneous neuronal activity in the primary motor cortex (M1). The Authors hypothesize that these two stimulation methods will produce distinct patterns of modulation in spontaneous neuronal activity within the M1 region. Assessing neural activity is important for evaluating information processing, neuroplasticity, and cognitive performance in different health states and pathological conditions.
Respond to GC 1:
Thank you very much for your professional review comments. We have revised the paper according to each of your professional review comments. With your help, the quality of our paper has significantly improved.
Major comment 1. Participants underwent two 20-minute sessions of TI or tDCS stimulation. The order in which the different stimulation types were administered must be specified or whether they were randomized. Neuronal activity was assessed using both functional and structural imaging methods.
Respond to MC 1:
We sincerely appreciate your careful review. In our study, we implemented a randomized crossover design with counterbalanced stimulation sequences to mitigate order effects. This was not explicitly detailed in the original manuscript. In the revised version, we have added the detail of counterbalanced.
“To prevent order effects, the participants were randomly assigned to receive the two different types of stimulation by an experimenter who not involved in data processing.” (P3, line 117-119)
Major comment 2. The article is written clearly, has a clear structure. The conclusions correspond to the results. There are minor grammatical errors. A grammar check is necessary. Pay attention to sentence structure and spelling. You should also check for abbreviations.
Respond to MC 2:
Thank you for your suggestions regarding grammatical inconsistencies, repetitive expressions, and formatting issues in the manuscript. We are grateful for your attention to these aspects and fully agree with your opinions. We have checked the grammar by the “paperpal” throughout the text, eliminated repetitive expressions, and corrected formatting issues. Through these revisions, the overall quality of the manuscript has been significantly improved.

Round 2
Reviewer 2 Report
Comments and Suggestions for Authors
I have no more comments.